# CCR4, a RNA decay factor, is hijacked by a plant cytorhabdovirus phosphoprotein to facilitate virus replication

Zhen-Jia Zhang[1], Qiang Gao[1], Xiao-Dong Fang[1], Zhi-Hang Ding[1], Dong-Min Gao[1], Wen-Ya Xu[1], Qing Cao[1], Ji-Hui Qiao[1], Yi-Zhou Yang[1], Chenggui Han[2], Ying Wang[2], Xuefeng Yuan[3], Dawei Li[1], Xian-Bing Wang[1]*

[1]State Key Laboratory of Agro-Biotechnology, College of Biological Sciences, China Agricultural University, Beijing, China; [2]College of Plant Protection, China Agricultural University, Beijing, China; [3]Department of Plant Pathology, College of Plant Protection, Shandong Agricultural University, Shandong Province Key Laboratory of Agricultural Microbiology, Tai'an, China

**Abstract** Carbon catabolite repression 4 (CCR4) is a conserved mRNA deadenylase regulating posttranscriptional gene expression. However, regulation of CCR4 in virus infections is less understood. Here, we characterized a pro-viral role of CCR4 in replication of a plant cytorhabdovirus, *Barley yellow striate mosaic virus* (BYSMV). The barley (*Hordeum vulgare*) CCR4 protein (HvCCR4) was identified to interact with the BYSMV phosphoprotein (P). The BYSMV P protein recruited HvCCR4 from processing bodies (PBs) into viroplasm-like bodies. Overexpression of HvCCR4 promoted BYSMV replication in plants. Conversely, knockdown of the small brown planthopper CCR4 inhibited viral accumulation in the insect vector. Biochemistry experiments revealed that HvCCR4 was recruited into N–RNA complexes by the BYSMV P protein and triggered turnover of N-bound cellular mRNAs, thereby releasing RNA-free N protein to bind viral genomic RNA for optimal viral replication. Our results demonstrate that the co-opted CCR4-mediated RNA decay facilitates cytorhabdovirus replication in plants and insects.

*For correspondence:
wangxianbing@cau.edu.cn

**Competing interests:** The authors declare that no competing interests exist.

## Introduction

In eukaryotic cells, mRNA levels determined by biosynthesis and turnover tightly regulate protein production in response to cellular environment changes (*Collart, 2016*; *Yu et al., 2019*). The mRNA poly(A) tail is important for post-transcriptional regulation by affecting mRNA stability and translation (*Wiederhold and Passmore, 2010*). Shortening of mRNA poly (A) tails, a process known as deadenylation, is the rate-limiting step in the mRNA decay process (*Garneau et al., 2007*; *Parker and Sheth, 2007*; *Houseley and Tollervey, 2009*). In eukaryotes, mRNA deadenylation is primarily mediated by the CCR4-NOT complex in RNA processing bodies (PBs) that are dynamic cytoplasmic structures containing silenced mRNAs for degradation or translation repression (*Parker and Sheth, 2007*; *Beckham and Parker, 2008*; *Xu and Chua, 2011*; *Miller and Reese, 2012*; *Chen and Shyu, 2013*; *McCormick and Khaperskyy, 2017*). Accumulating evidence shows that the CCR4-NOT complex is recruited to some specific mRNAs for deadenylation by RNA-binding proteins (RBPs) such as Pumilio, Roquin, and Tristetraprolin (*Leppek et al., 2013*; *Wahle and Winkler, 2013*; *Arae et al., 2019*; *Webster et al., 2019*).

In the CCR4-NOT complex, carbon catabolite repression 4 (CCR4) and CCR4 associated factor (CAF1) are two deadenylases responsible for removing mRNA poly(A) tails (*Collart, 2016*). CAF1 interacts directly with the leucine-rich repeat (LRR) domain of CCR4 and the MIF4G domain of NOT1, and hence links CCR4 to the NOT1 scaffold protein of the CCR4-NOT complex (*Zuo and*

*Deutscher, 2001*; *Basquin et al., 2012*). While, plant CCR4 proteins do not contain LRR domains at their N termini (*Dupressoir et al., 2001*; *Chou et al., 2017*). Instead, the N termini of plant CCR4 orthologues contain a zf-MYND-like domain that interacts with plant CAF1 proteins in the CCR4-NOT complex (*Chou et al., 2017*). Cellular mRNA decay is usually manipulated by viruses to complete viral life cycles due to limited functional proteins in viruses (*Ariumi et al., 2007*; *Beckham and Parker, 2008*; *Dougherty et al., 2011*; *McCormick and Khaperskyy, 2017*; *Guo et al., 2018*). Currently, regulation of CCR4 and its deadenylase activity in viral infections is not well understood.

Rhabdoviruses are negative-stranded RNA viruses that infect a wide range of organisms including plants, vertebrates, and invertebrates (*Ammar et al., 2009*; *Mann and Dietzgen, 2014*; *Dietzgen et al., 2016*). Rhabdoviruses share similar genome organizations encoding five structural proteins, including the nucleoprotein (N), phosphoprotein (P), matrix protein (M), glycoprotein (G), and large polymerase (L) proteins in a conserved order 3′–N–P–M–G–L–5′. In addition, a number of accessory proteins are encoded in the overprinted, overlapped, and interspersed regions of the structural protein genes (*Walker et al., 2011*). The three core proteins, N, P, L proteins, and the genomic RNA molecule compose viral capsid complex (NC) that functions in elegantly regulated viral replication and transcription cycles (*Ivanov et al., 2011*). The N protein entirely encapsidates the gRNA and antigenomic (ag) RNA to form N-RNA complexes that serves as templates for replication and transcription (*Ivanov et al., 2011*). The multiple functional P protein chaperones nascent RNA-free N ($N^0$) by forming an $N^0$–P complex that prevent $N^0$ from binding nonspecifically to cellular RNAs, and attaches L protein polymerase complexes to the N-RNA complex (*Ivanov et al., 2011*; *Leyrat et al., 2011a*). Accumulating evidence provides preliminary glimpses of the mechanisms whereby P acts multiple roles in viral RNA synthesis (*Ivanov et al., 2011*; *Leyrat et al., 2011a*), but numerous unanswered questions about the molecular mechanisms still need to be explored. For example, it is not well understood how P binds to $N^0$ to prevent binding to cellular mRNAs and facilitate specific viral gRNA and agRNA interactions. In particular, it remains to be determined whether cellular factors are involved into these processes.

*Barley yellow striate mosaic virus* (BYSMV), a member of *Cytorhabdovirus* genus, infects cereal crops and causes yield losses worldwide. In 2015, BYSMV was first reported in wheat fields of northern China (*Di et al., 2014*). BYSMV is obligately transmitted by the small brown planthopper (SBPH) in a persistent propagative manner (*Cao et al., 2018*). The complete BYSMV genome encodes ten proteins in the order of 3′–N–P–P3–P4/P5–P6–M–G–P9–L–5′ (*Yan et al., 2015*). Recently, we successfully established the BYSMV minireplicon system and rescued BYSMV from full length cDNA clones (*Fang et al., 2019*; *Gao et al., 2019*). BYSMV reverse genetic systems allow us to begin to dissect interactions between BYSMV and its plant hosts and insect vectors and to engineer versatile delivery platforms in monocot plants and planthoppers (*Fang et al., 2019*; *Gao et al., 2019*). Here, we demonstrate that the BYSMV P protein interacts directly with CCR4 and recruits the protein into viroplasm-like bodies to improve viral RNA replication. We also show that the P-recruited CCR4 proteins are responsible for turnover of the N-bound cellular mRNAs. Our results suggest a positive role of CCR4 in cytorhabdovirus infection cycles and explain how the P protein binds to $N^0$ to prevent binding of cellular mRNAs.

## Results

### BYSMV P interacts with barley HvCCR4 in vitro and in vivo

Rhabodovirus P proteins have essential roles in viral genomic RNA replication and mRNA transcription (*Ivanov et al., 2011*). To fully understand rhabdovirus P functions in vivo, we screened host factors interacting with the BYSMV P protein and investigated their functions during infections. To this end, we expressed GST-P in *E. coli* and purified this protein to serve as a bait to immunoprecipitate (IP) P protein interacting proteins from infected barley leaves. The IP products were separated in SDS-PAGE gels and analyzed by liquid chromatography-tandem mass spectrometry (LC-MS/MS), and a barley CCR4 fragment was identified amongst the IP products (*Supplementary file 1*). Then, the sequences of CCR4 orthologues of *Hordeum vulgare* (HvCCR4) and *Laodelphax striatellus* (LsCCR4) were obtained from NCBI protein sequence databases (*Figure 1—figure supplement 1*). Both plant and animal CCR4 proteins contain a conserved EEP domain in their C-termini (*Figure 1A*; *Figure 1—figure supplements 1* and *2*). Plant CCR4 orthologues harbor a zf-MYND-like domain at

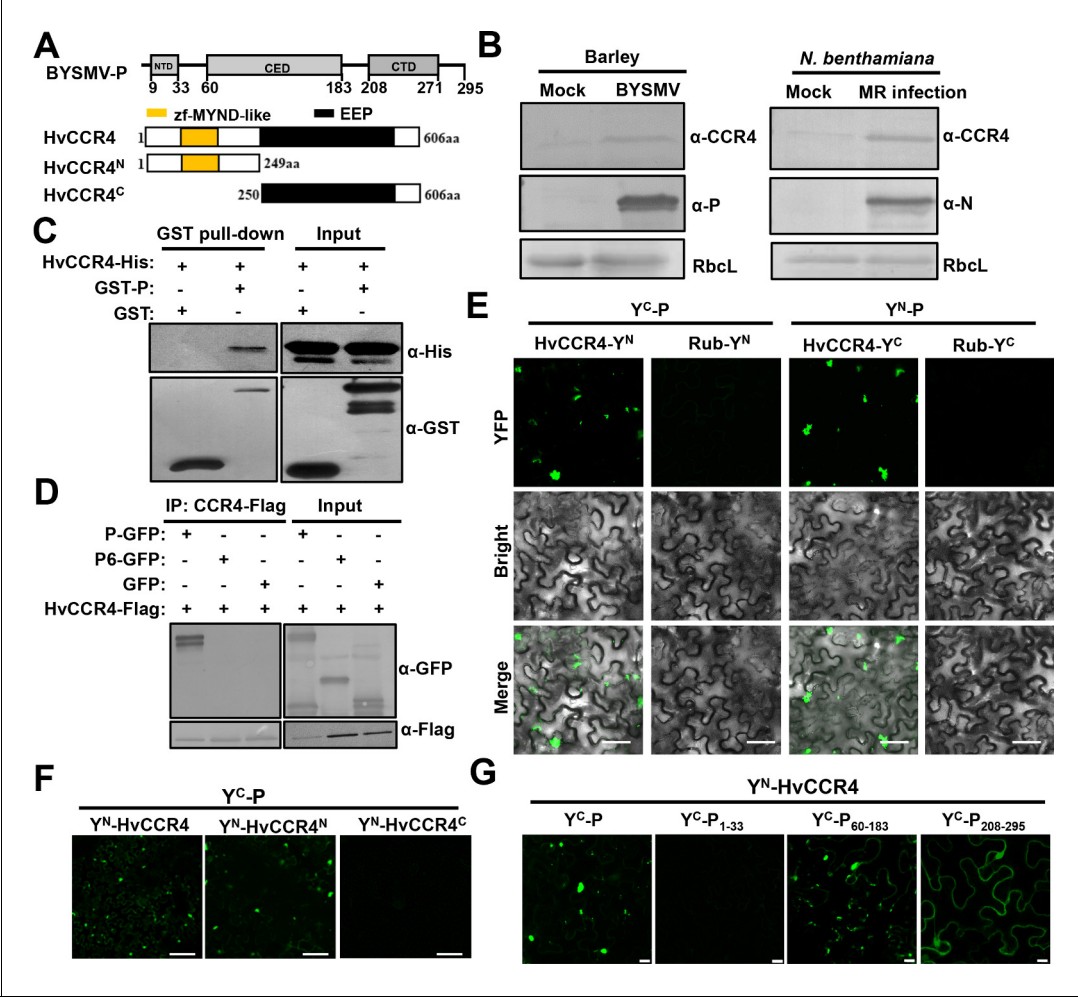

**Figure 1.** The barley HvCCR4 protein interacts with the BYSMV P protein in vitro and in vivo. (**A**) Modular organization of BYSMV P, HvCCR4 and schematic presentation of deletion mutants. The BYSMV P protein consists of three structural domains, including an N terminal domain (NTD), a central domain (CED), and a C terminal domain (CTD). The yellow and black boxes indicate the HvCCR4 zf-MYND-like and EEP nuclease domains. (**B**) Western blotting analyses showing accumulation of the CCR4 (α-CCR4), P (α-P) and N (α-N) proteins in BYSMV-infected barley plants, BYSMV-MR-infected *N. benthamiana* leaves, and mock inoculated plants. Rubisco complex large subunit (RbcL) detected by Stain-Free technology was used as a loading control. (**C**) GST pull-down assays showing HvCCR4–P interactions in vitro. His-tagged HvCCR4 was incubated with GST-tagged P or GST and immunoprecipitated with glutathione-Sepharose beads. The pull down and input proteins were detected by western blotting assays with anti-GST (α-GST) and anti-His (α-His) antibodies. (**D**) Co-IP analysis of the interaction between P-GFP and HvCCR4-Flag in vivo. *N. benthamiana* leaves were agroinfiltrated with *A. tumefaciens* cells expressing various proteins as indicated. At three dpi, leaf extracts were incubated with anti-Flag beads, and then the IP and input proteins were analyzed by western blotting with anti-GFP (α-GFP) and anti-Flag (α-GFP) antibodies. (**E**) BiFC analysis of the HvCCR4–P interaction in epidermal cells of *N. benthamiana* leaves infiltrated with *Agrobacterium* strains expressing proteins tagged with the $Y^N$ or $Y^C$ halves of YFP. Rubisco protein (Rub) served as a negative control. Bar = 50 μm. (**F**) BiFC analysis of BYSMV P interactions with the $HvCCR4^N$ or $HvCCR4^C$ domains. Bar = 50 μm. (**G**) BiFC analysis of the interaction between HvCCR4 and BYSMV P domains indicated in panel A. Bar = 10 μm. In (**E**), (**F**), and (**G**), images were taken at three dpi with a Leica laser scanning microscope.

The online version of this article includes the following figure supplement(s) for figure 1:

**Figure supplement 1.** Phylogenetic tree of plant and animal CCR4 orthologues.

**Figure supplement 2.** Sequence alignments of the zf-MYND-like, LRR, and EEP domains of various CCR4 protein.

**Figure supplement 3.** qRT-PCR analysis of the expression of CCR4 in BYSMV-infected and mock-treated barley at 15 dpi or planthoppers at seven dpi.

**Figure supplement 4.** BiFC assays showing negative Rub controls with HvCCR4 and LsCCR4.

**Figure supplement 5.** Time-lapse confocal images of $Y^N$-P and $Y^C$-HvCCR4 in *N*.

**Figure supplement 6.** CCR4 does not affect self-interaction of BYSMV P in vitro and in vivo.

**Figure supplement 7.** BiFC assays of NCMV-P with HvCCR4 or LsCCR4.

the N termini (*Figure 1A*; *Figure 1—figure supplements 1* and *2*), while the N terminus of LsCCR4 contains an LRR domain that is highly conserved in yeast and mammalian CCR4 orthologues (*Figure 1—figure supplements 1* and *2*). These results are well consistent with the classification of CCR4 proteins in previous studies (*Chou et al., 2017*).

To experimentally examine the response of CCR4 to BYSMV infections, CCR4 protein levels were detected in infected barley plants and BYSMV minireplicon infection in *N. benthamiana* (*Fang et al., 2019*; *Gao et al., 2019*). Western blotting shows that CCR4 protein accumulation increased in BYSMV-infected barley leaves at 15 dpi (*Figure 1B*, left panel) and in BYSMV-minireplicon-infected *N. benthamiana* plants at five dpi (*Figure 1B*, right panels) compared with mock plants. However, qRT-PCR assays revealed that accumulation of the CCR4 mRNA had no significant difference (p<0.01) and even a little reduction in BYSMV-infected than in mock-treated plants (*Figure 1—figure supplement 3*). These results demonstrate that BYSMV infections improve accumulation of host CCR4 protein rather than its mRNA.

To determine direct interactions between the BYSMV P protein and HvCCR4 in vitro, HvCCR4 fused with a 6 × His tag (CCR4-His) and P fused with a GST tag (GST-P) were purified from *E. coli*. GST pull-down assays revealed that HvCCR4-His interacted with GST-P in vitro, but not with the GST control (*Figure 1C*). To further examine the P–HvCCR4 interaction in vivo by coimmunoprecipitation (Co-IP) assays, HvCCR4-Flag was co-expressed with GFP, P-GFP, or P6-GFP in *N. benthamiana* leaves by agroinfiltration. Note that the BYSMV P6 protein served as a negative control in Co-IP assays. At two dpi, Co-IP assays with agroinfiltrated *N. benthamiana* leaves show that HvCCR4-Flag could efficiently immunoprecipitate P-GFP, but not GFP or P6-GFP (*Figure 1D*).

We alse carried out BiFC assays to determine whether BYSMV P associates with HvCCR4 in living cells. For these assays, HvCCR4 or BYSMV P was fused to the C ($Y^C$) or N ($Y^N$) halves of sYFP, and co-expressed in *N. benthamiana* leaves. BiFC fluorescence was not observed in the negative control combinations of either $Y^C$-P/Rub-$Y^N$ or $Y^N$-P/Rub-$Y^C$ (*Figure 1—figure supplement 4*). In contrast, co-expression of $Y^C$-P/$Y^N$-HvCCR4 and $Y^N$-P/$Y^C$-HvCCR4 resulted in production of YFP fluorescent punctate granules in the cytoplasm (*Figure 1E*). Notably, these fluorescent punctate granules moved rapidly throughout the cytoplasm (*Video 1*). Furthermore, when treated with the actin-depolymerizing agent 10 μM LatB, movement of the BiFC bodies of $Y^N$-P/$Y^C$-HvCCR4 was abolished in contrast to the highly dynamic movement in the DMSO control (*Figure 1—figure supplement 5*, *Videos 2* and *3*). These results, together with our previous results (*Fang et al., 2019*), reveal that the BYSMV P protein and host HvCCR4 protein are components of mobile inclusion bodies that traffic along the actin/endoplasmic reticulum network.

The BYSMV P protein contains three structural domains consisting of an N-terminal domain (P$_{NTD}$, aa 9–33), a central domain (P$_{CED}$, aa 60–183), and a C-terminal domain (P$_{CTD}$, aa 208–270) (*Figure 1A*). The HvCCR4 protein harbors an N-terminal zf-MYND-like domain and a C-terminal EEP domain (*Figure 1A*). To determine the interaction domains of these proteins, the N (aa 1–149) and C termini (aa 250–606) of HvCCR4 were each fused to $Y^N$, and the three BYSMV P domains P$^{1-33}$ (aa 1–33), P$^{60-183}$ (aa 60–183), and P$^{208-295}$ (aa 208–295) were individually fused to $Y^C$ for BiFC assays. Coexpression of $Y^C$-P and $Y^N$-HvCCR4$^N$ resulted in production of BiFC fluorescent punctate granules, whereas $Y^N$-HvCCR4$^C$ did not associate with $Y^C$-P (*Figure 1F*). Both P$^{60-183}$ and P$^{208-295}$ associated with CCR4, but only P$^{60-183}$ and CCR4 the association formed punctate granules similar to those of wild type BYSMV P (*Figure 1G*). Moreover, we determined that CCR4 did not affect self-interaction of BYSMV P in vitro and in vivo using competitive GST pull-down and BiFC assays

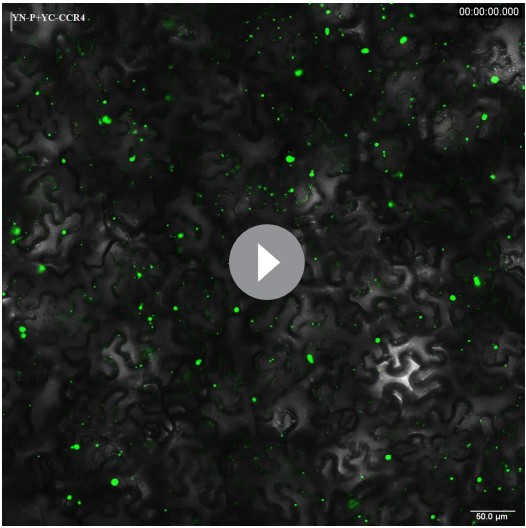

**Video 1.** Motile cytoplasmic BiFC bodies of $Y^N$-P and $Y^C$-CCR4 in the epidermal cells of agroinfiltrated *N. benthamiana* leaves. Images were taken at three dpi with a Leica laser scanning microscope. Bar = 50 μm.
https://elifesciences.org/articles/53753#video1

(*Figure 1—figure supplement 6*). Collectively, our results indicate that BYSMV P interacts directly with HvCCR4 in vitro and in vivo.

Furthermore, the N-terminal zf-MYND-like domain of HvCCR4 and the central BYSMV P domain ($P^{60-183}$) are responsible for the P–HvCCR4 body formation. Besides, we found that the P protein of northern cereal mosaic virus (NCMV), a close related cytorhabdovirus, interacts with the N-terminal zf-MYND-like domain of HvCCR4 using BiFC assays (*Figure 1—figure supplement 7*). These results demonstrate that the P–CCR4 interaction is probably a common feature in cytorhabdoviruses.

## BYSMV P hijacks HvCCR4 from PBs into viroplasm-like bodies

Arabidopsis and rice CCR4 are localized in processing bodies (PBs) that are foci for mRNA turnover in the cytoplasm (*Beckham and Parker, 2008*; *Chen and Shyu, 2013*; *Suzuki et al., 2015*; *Chou et al., 2017*). The decapping protein 1 (DCP1) is a well-known PB protein, so we fused this protein to the N-terminus of mCherry (DCP1-mCherry) to provide a PB localization marker (*Kumakura et al., 2009*; *Xu and Chua, 2011*). Then, HvCCR4-GFP and DCP1-mCherry were transiently co-expressed in *N. benthamiana* leaves by agroinfiltration. As expected, HvCCR4-GFP and DCP1-mCherry co-localized in punctate cytoplasmic PBs (*Figure 2A*). Interestingly, GFP-P and HvCCR4-mCherry fluorescence were associated with trafficking punctate granules in cytoplasm (*Figure 2B* and *Video 4*), but the GFP-P granules were distinct from but adjacent to the DCP1-mCherry labeled marker PBs (*Figure 2C*, upper panel). Interestingly, the GFP-P bodies and DCP1-mCherry bodies trafficked and were adjacent with each other in cytoplasm (*Figure 2C*, bottom panel and *Video 5*), implying that some undefined elements may have functioned to associate the two bodies. In contrast, DCP1-mCherry bodies could not be adjacent with GFP-$P_{1-207}$ that cannot form inclusion bodies as described in our previous study (*Figure 2—figure supplement 1*; *Fang et al., 2019*).

To further investigate their co-localization, CFP-P, HvCCR4-GFP, and DCP1-mCherry were coexpressed in *N. benthamiana* leaves, and the results revealed that HvCCR4-GFP and CFP-P fully overlapped in punctate granules that were adjacent to DCP1-mCherry-labelled PBs (*Figure 2D*). In contrast, HvCCR4-GFP and DCP1-mCherry fully overlapped in P bodies when co-expressed with the BYSMV CFP-N protein (*Figure 2E*). Quantitative analyses revealed that fifty-eight among sixty HvCCR4-GFP bodies were adjacent to DCP1-mCherry bodies, while only two HvCCR4-GFP bodies were overlapped with DCP1-mCherry bodies in the presence of the CFP-P protein (*Figure 2—figure supplement 2*, left panel). However, most of HvCCR4-GFP bodies (fifty-six among sixty in total) were overlapped with DCP1-mCherry bodies in the presence of the CFP-N protein (*Figure 2—figure supplement 2*, right panel). These results indicate that the BYSMV P protein, rather than the N protein, hijacks host HvCCR4 from PBs in cytoplasm.

Recently, we demonstrated that BYSMV P serves as a dynamic tethering protein to recruit the N and L proteins into viroplasm-like bodies that traffic along the ER/actin network (*Fang et al., 2019*). Therefore, the colocalization results above prompted us to investigate whether the BYSMV P protein also hijacks HvCCR4 into viroplasm-like bodies that contain the BYSMV N, P, and L proteins. To this end, $Y^N$-P and $Y^C$-L were co-expressed with CFP-N and HvCCR4-mCherry in *N. benthamiana* leaves by agroinfiltration. As expected, the $Y^N$-P/$Y^C$-L fluorescence colocalized with CFP-N and HvCCR4-mCherry (*Figure 3A*, upper panel). In contrast, free mCherry distributed evenly in nucleus and cytoplasm, but not co-localized with $Y^N$-P and $Y^C$-L, and CFP-N (*Figure 3A*, bottom panel). Moreover, HvCCR4-GFP was recruited to the cytoplasmic granules together with CFP-N and L-mCherry in the presence of the BYSMV P protein (*Figure 3B*, upper panel), and these bodies moved rapidly throughout the cytoplasm (*Video 6*). However, without expression of BYSMV P, CFP-N and L-mCherry could not co-localized with the HvCCR4-GFP protein (*Figure 3B*, bottom panel).

To more directly confirm the presence of HvCCR4 in the viroplasms, we agroinfiltrated plasmid mixtures in *N. benthamiana* leaves for expression of BYSMV minigenome (agMR), viral suppressor of RNA silencing (VSRs), N-Flag, L-Myc, P, along with HvCCR4-GFP or GFP. At five dpi, total proteins from infiltrated leaves were immunoprecipitated with anti-Flag affinity beads and analyzed by western blotting analysis. Co-IP assays showed that HvCCR4-GFP, but not GFP, coprecipitated with N-Flag, L-Myc, and P (*Figure 3C*, compare lanes 1 and 2). In the absence of P protein expression, N-Flag did not coprecipitate with L-Myc or HvCCR4-GFP (*Figure 3C*, compare lane 3), indicating that the P protein is a central tethering protein for the N, L and HvCCR4 protein complexes.

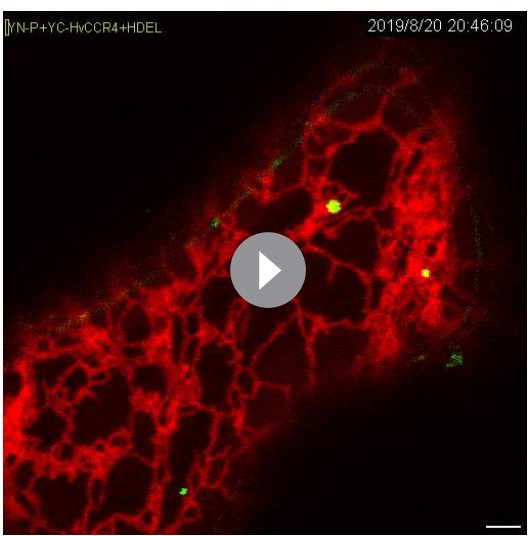

**Video 2.** Intracellular movement of $Y^N$-P and $Y^C$-CCR4 inclusion bodies in epidermal cells of agroinfiltrated *N. benthamiana* leaves expressing mCherry-HDEL after DMSO treatment. Scale bar = 2 μm.
https://elifesciences.org/articles/53753#video2

Collectively, these results indicate that BYSMV P acting as a dynamic tethering protein efficiently recruit both viral proteins and host HvCCR4 into viroplasm-like bodies, and suggest that the host HvCCR4 protein functions in virus replication and/or transcription.

## HvCCR4 deadenylase activity enhances BYSMV minigenome RNA replication

CCR4 is a major cytoplasmic deadenylase triggering mRNA deadenylation and decay (*Collart, 2016*). Rhabdoviruses transcribe capped and polyadenylated mRNAs that are potentially targeted by host CCR4, which may be involved in defense against infection. To verify this hypothesis, we overexpressed HvCCR4 and various derivatives in *N. benthamiana* leaves with BYSMV minireplicon system including the antigenomic minireplicon (agMR), the N, P, L proteins, and VSRs (*Figure 4A*). In the agMR construct, GFP and RFP are flanked by the BYSMV N- and C-terminal sequences and act as fluorescent reporters to monitor agMR infections (*Fang et al., 2019*). Compared to the empty vector (EV), overexpression of HvCCR4 increased the numbers of RFP fluorescence foci at five dpi (*Figure 4B*), but the HvCCR4$^N$ or HvCCR4$^C$ terminal derivatives did not increase the number of fluorescence foci (*Figure 4B*). The conserved Asn260 and Glu305 residues of HvCCR4 are essential for deadenylase activity (*Figure 1—figure supplement 2C*; *Chou et al., 2017*), and overexpression of alanine-substituted mutant (N260A/E305A, designated CCR4$^{mEEP}$) also did not increase the fluorescence foci, compared with empty vector (*Figure 4B*, right panel). These results suggest that that HvCCR4 deadenylase activity substantially increases BYSMV agMR infections.

Western blotting was performed to evaluate accumulation of BYSMV agMR RFP. Consistent with the numbers of fluorescence foci in infiltrated cells, RFP accumulated to higher levels in HvCCR4 coinfiltrated tissues compared with those of EV, HvCCR$^N$, HvCCR$^C$, and HvCCR4$^{mEEP}$ (*Figure 4C*). In contrast, overexpression of HvCCR4 had negligible effects on accumulation of BYSMV N and P (*Figure 4C*), indicating improvement of HvCCR4 in MR infections was independent of N and P accumulation. Quantitative RT-PCR (qRT-PCR) analysis was further performed to determine levels of agMR replication and transcription. The full-length MR RNA abundance representative of MR replication increased to 2.28-fold in the HvCCR4 overexpression samples compared with the EV controls (*Figure 4D*). In contrast, HvCCR$^N$, HvCCR$^C$, and HvCCR4$^{mEEP}$ co-expression had negligible effects on MR replication (*Figure 4D*). In addition, we evaluated transcription activities by normalizing the RFP mRNA levels relative to the gMR template, showing that all the samples

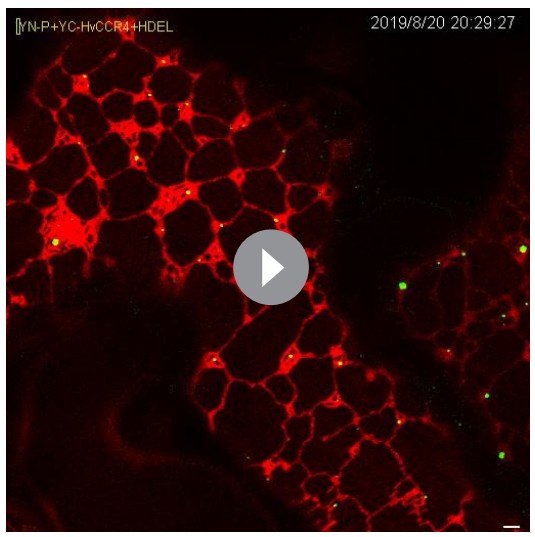

**Video 3.** Intracellular movement of $Y^N$-P and $Y^C$-CCR4 inclusion bodies in epidermal cells of agroinfiltrated *N. benthamiana* leaves expressing mCherry-HDEL with LatB (10 μM) treatment. Scale bar = 2 μm.
https://elifesciences.org/articles/53753#video3

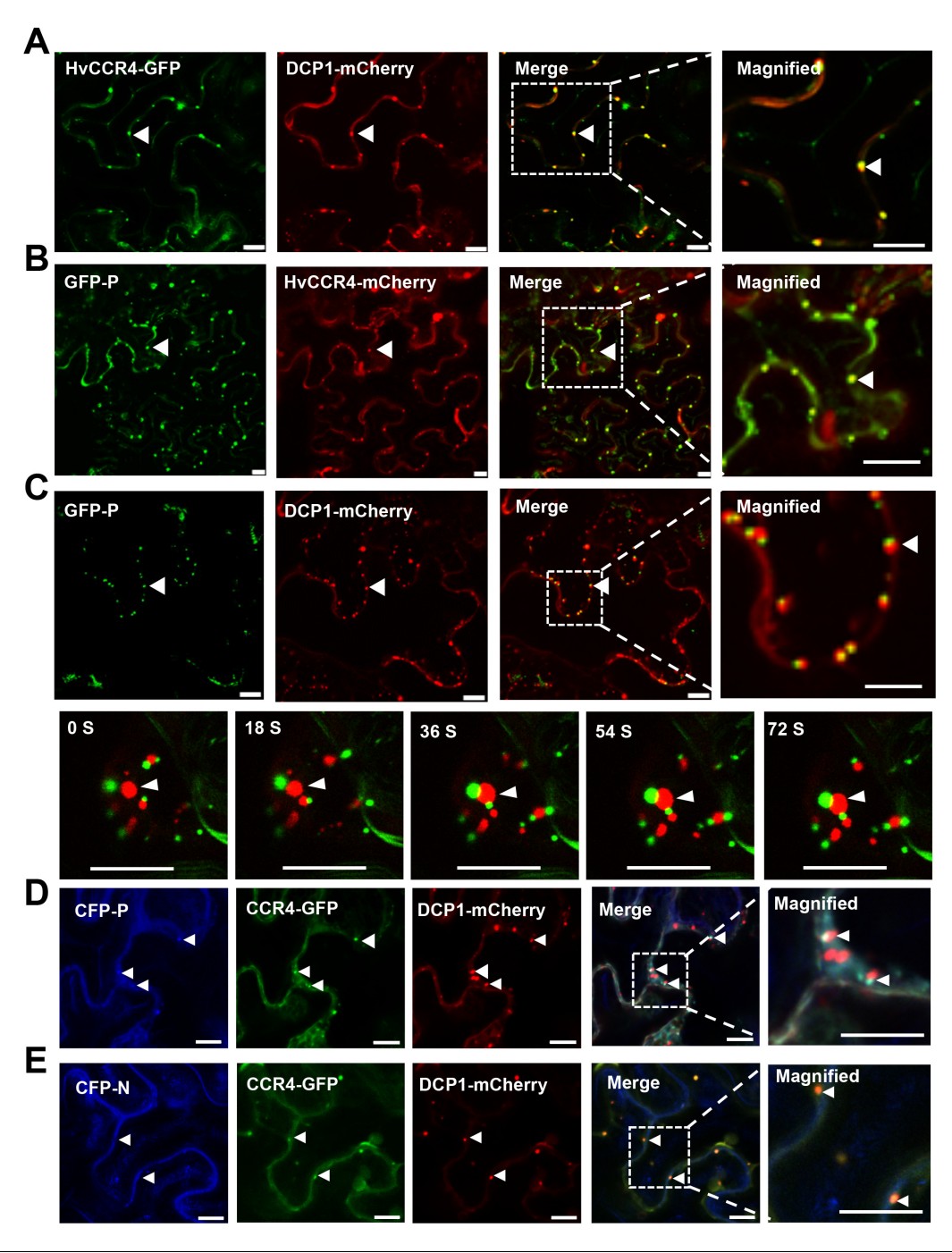

**Figure 2.** BYSMV P sequestration of HvCCR4 from the PBs. (**A–C**) Confocal micrographs showing subcellular co-localization of HvCCR4-GFP and DCP1-mCherry (**A**), GFP-P and HvCCR4-mCherry (**B**), GFP-P and DCP1-mCherry (**C**) co-expressed in *N. benthamiana* epidermal cells. In the bottom panel (**C**), time-lapse confocal images of GFP-P and DCP1-mCherry. Arrow heads indicate association of different bodies. Magnified images (in white boxes) are shown in the right panels. Bar = 10 μm. (**D and E**) Transient co-expression of HvCCR4-GFP and DCP1-mCherry with CFP-P (**D**) or CFP-N (**E**) were monitored at three dpi. Bar = 10 μm.

The online version of this article includes the following figure supplement(s) for figure 2:

**Figure supplement 1.** Co-localization of GFP-P or GFP-P$_{1-207}$ with DCP1-mCherry.

**Figure supplement 2.** Quantitative data showing localization of CCR4-GFP and DCP1-mCherry bodies in the absence and presence of P protein.

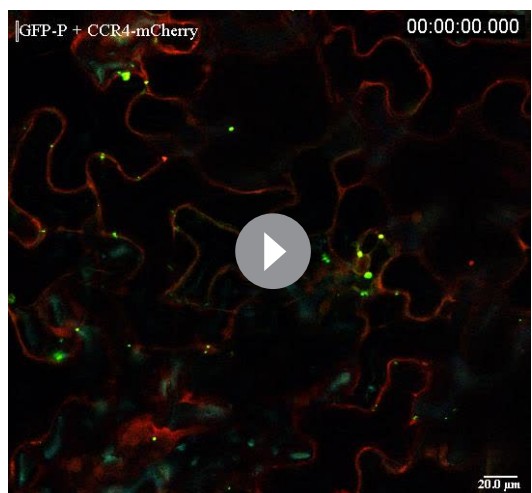

**Video 4.** High mobility of GFP-P and CCR4-mCherry colocalized bodies in epidermal cells of agroinfiltrated *N. benthamiana* leaves. Scale bar = 20 µm.
https://elifesciences.org/articles/53753#video4

had similar transcription activities (*Figure 4E*). Collectively, these results indicate that HvCCR4 promotes minigenome replication, but not virus transcription.

## HvCAF1 has a negative effect on BYSMV minigenome replication

In eukaryote organisms, the highly conserved CCR4-NOT complex contains two deadenylases, CCR4 and CAF1, that provide major contributions to mRNA deadenylation and turnover in PBs (*Collart, 2016*). In yeast, CAF1 binds both the NOT1 MIF4G (middle portion of eIF4G) and the CCR4 LRR domain, thereby bridging NOT1 and CCR4 (*Basquin et al., 2012*). Unlike the LRR domain of animal and yeast CCR orthologues, plant CCR4 proteins contain Mynd-like domains that interact with the PXLXP motif at the N-terminus of CAF1 (*Figure 5A*; *Chou et al., 2017*). We first cloned the full-length cDNA sequence of barley CAF1 and analyzed the amino acid sequence of HvCAF1 (*Figure 5—figure supplement 1A*), and found that HvCAF1 contains a classical PXLXP motif that interacts directly with the CCR4 N terminus (*Figure 5—figure supplement 1B*). To confirm the interaction of HvCCR4 and HvCAF1, the HvCAF1 ORFs were fused to $Y^C$ (HvCAF1-$Y^C$), and then coexpressed with HvCCR4$^N$-$Y^N$ or HvCCR4$^C$-$Y^N$ for BiFC assays. As expected, BiFC analysis shows that HvCAF1 interacts with the HvCCR4 N terminus, but has compromised affinity with the HvCCR4 C terminus (*Figure 5—figure supplement 2A*). In addition, our results show that BYSMV P does not interact with HvCAF1 in BiFC assays (*Figure 5—figure supplement 2B*). It should be noted that both BYSMV P and HvCAF1 interact with the Mynd-like domain of HvCCR4.

We next determined the function of CAF1 in BYSMV minigenome replication. As described above, HvCCR4 overexpression increased the numbers of RFP fluorescence foci in agroinfiltrated tissues (*Figure 5B*). In contrast, overexpression of HvCAF1 resulted in substantially decreased cell numbers of fluorescence foci at five dpi (*Figure 5B*). Western blotting analysis consistently showed that RFP accumulated to a much lower level in HvCAF1 overexpressed leaves, but accumulated to a higher level in HvCCR4 overexpressed leaves compared with empty vector infiltration (*Figure 5C*). Therefore, HvCAF1 negatively affects BYSMV minigenome infections, in contrast to HvCCR4 enhancement.

Because both BYSMV P and HvCAF1 interact with the Mynd-like domain of HvCCR4, we hypothesized that HvCAF1 interferes competitively with the HvCCR4 interactions with BYSMV P to reduce HvCCR4 enhancement in minigenome replication. Pull down assays showed that MBP-HvCAF1 interacts directly with HvCCR4-His, but not with P-His (*Figure 5D*). The HvCCR4-His and GST-P were incubated with increasing amounts of MBP-HvCAF1 (10, 20, and 30 µg) or MBP (30 µg), and western blotting

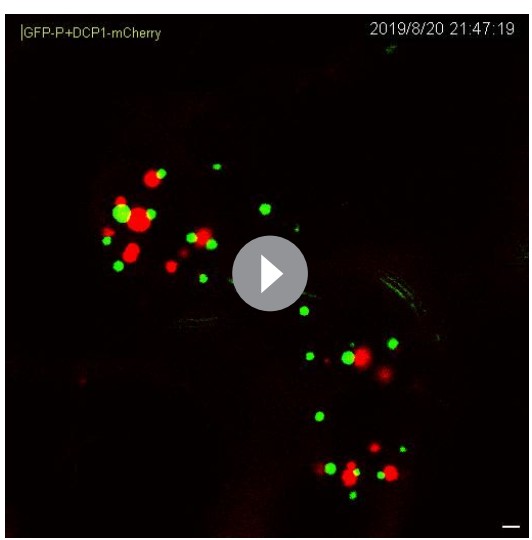

**Video 5.** High mobility of GFP-P bodies and DCP-mCherry bodies in epidermal cells of agroinfiltrated *N. benthamiana* leaves. Scale bar = 2 µm.
https://elifesciences.org/articles/53753#video5

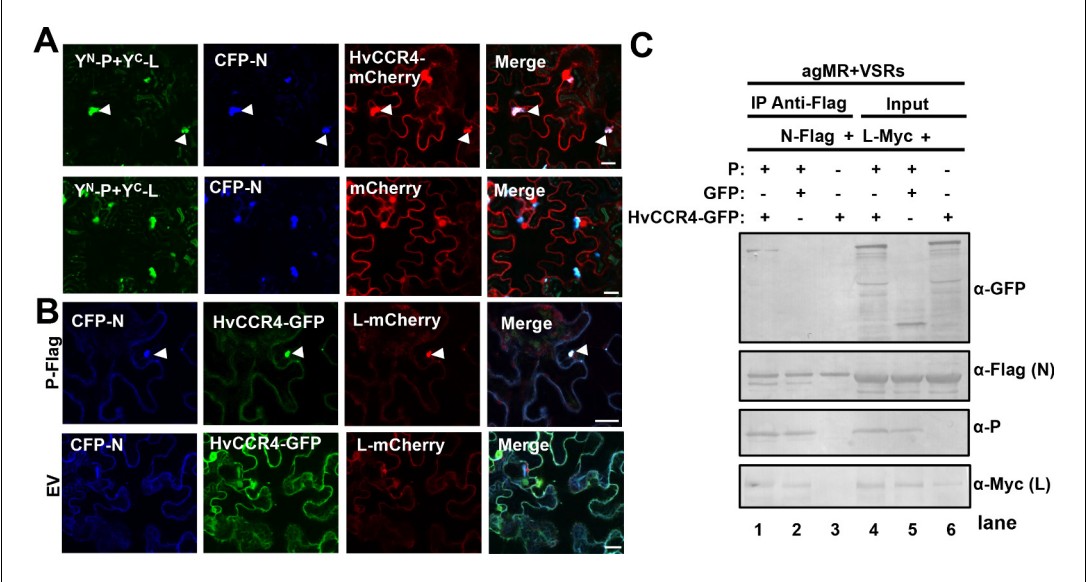

**Figure 3.** BYSMV P recruitment of HvCCR4 into viroplasm-like bodies. (A) Confocal micrographs showing subcellular localization of the $Y^N$-P and $Y^C$-L BiFC combination and CFP-N together with HvCCR4-mCherry or free mCherry in *N. benthamiana* leaves. Bar = 20 μm. (B) Confocal micrographs showing subcellular localization of CFP-N, HvCCR4-GFP, and L-mCherry in the presence of P-Flag or empty vector (EV). Bar = 20 μm. Representative images in panel A and B were taken at three dpi. Arrowheads indicated viroplasm-like bodies. (C) Co-IP analysis of the HvCCR4-GFP association with N-Flag, P, and L-Myc in vivo. *N. benthamiana* leaves were agroinfiltrated with *A. tumefaciens* cells expressing various proteins as indicated. At three dpi, leaf extracts were incubated with anti-Flag beads, and then the IP and input proteins were analyzed by western blotting with anti-GFP, -Flag, -P, and -Myc antibodies, respectively.

analysis revealed that reduced amount of HvCCR4-His was detected in GST pull-down products with increasing amount of MBP-HvCAF1 (*Figure 5E*, top panel, compare lanes 1–4). In contrast, free MBP had no effect on the P–CCR4 interactions (*Figure 5E*, lane 5), and free GST could not pull down HvCCR-His (*Figure 5E*, lane 6).

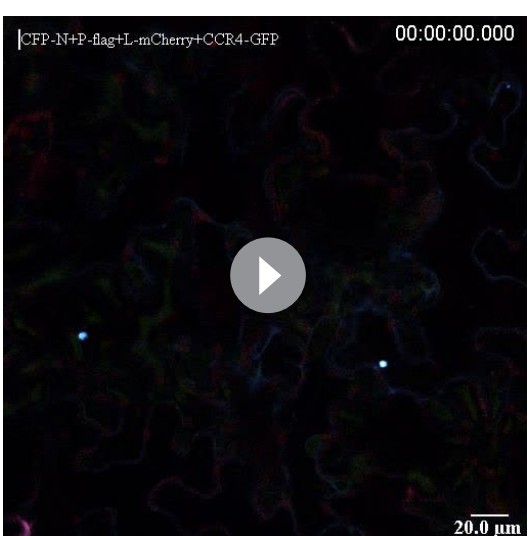

**Video 6.** A representative movie showing CFP-N, L-mCherry, and CCR4-GFP in epidermal cells of agroinfiltrated *N. benthamiana* leaves expressing BYSMV P. Scale bar = 20 μm.
https://elifesciences.org/articles/53753#video6

Competitive BiFC assays also revealed that overexpression of HvCAF1 inhibited the $Y^N$-P and $Y^C$-HvCCR4 interaction in a dose dependent manner (*Figure 5F* and *Figure 5—figure supplement 3*). As described above, $Y^N$-P and $Y^C$-HvCCR4 constituted YFP bodies co-localized with L-mCherry into viroplasm-like structures, which was almost disappear in the presence of HvCAF1 (*Figure 5—figure supplement 4*). These results indicate that overexpression of HvCAF1 negatively affect viroplasm-like structure formation.

Collectively, overexpressed HvCAF1 recruits HvCCR4 into CCR4-NOT1 complexes, which compromises the proviral functions of HvCCR4 in BYSMV replication. In contrast, the BYSMV P and HvCCR4 interactions probably interferes with the HvCAF1 and HvCCR4 binding, thereby inhibiting the formation of HvCCR4-NOT1 complexes. Thus, our results imply that HvCCR4 alone, rather than the HvCCR4-NOT1 complex, is hijacked by BYSMV P into viroplasm-like bodies for optimal viral replication.

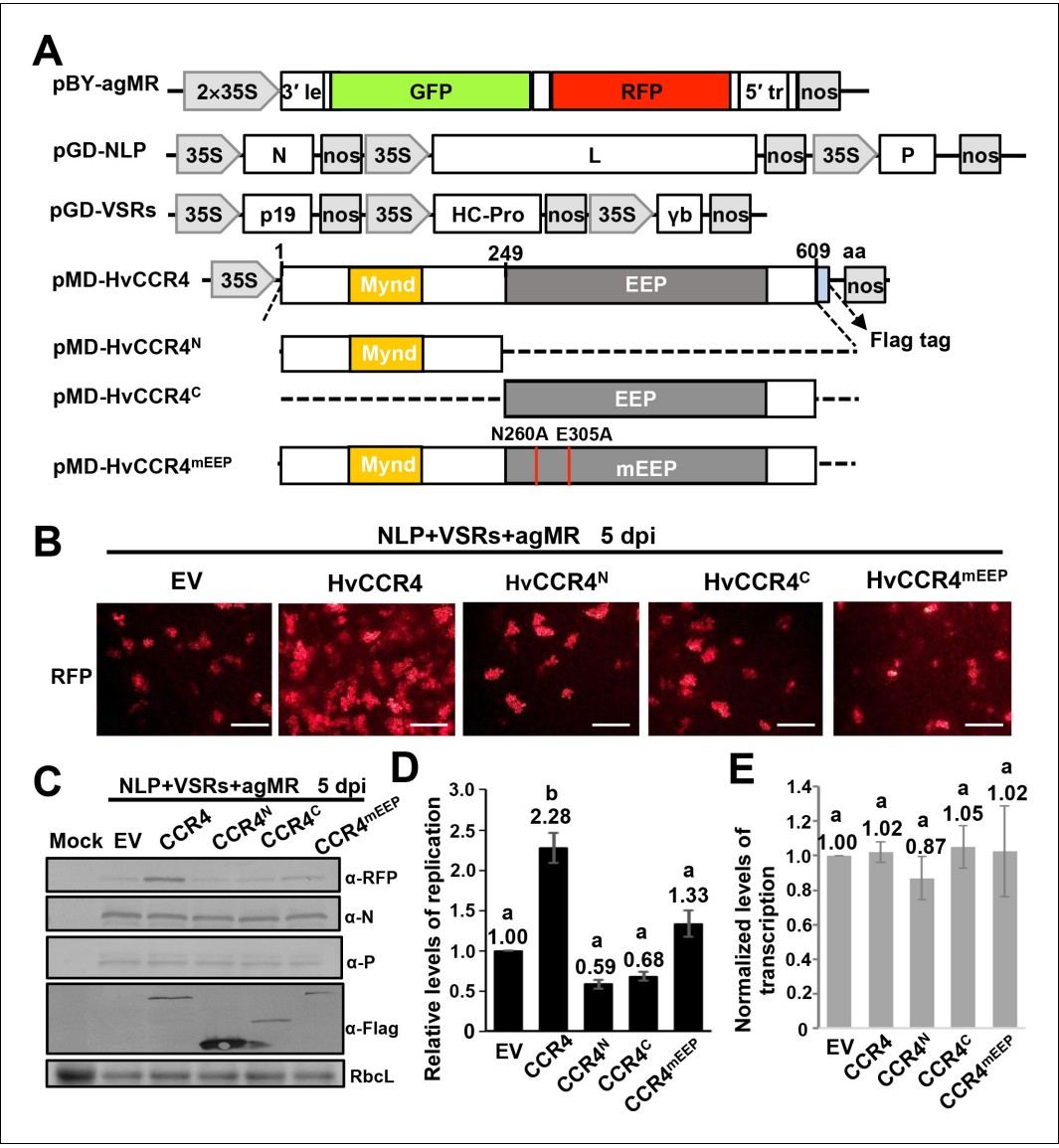

**Figure 4.** HvCCR4 enhancement of BYSMV minigenome replication. (**A**) Schematic diagrams of pBY-agMR, pGD-NLP, pGD-VSRs, pMD-HvCCR4, and derivative plasmids. Mixtures of these plasmids transiently express anti-genomic minireplicon RNA, BYSMV N, P, and L proteins, as well as suppressors of RNA silencing (VSRs) in *N. benthamiana* leaves. The HvCCR4, HvCCR4$^N$, HvCCR4$^C$ and HvCCR4$^{mEEP}$ (N260A/E305A) ORFs were cloned into pMDC32 for transient overexpression. (**B**) RFP foci in *N. benthamiana* leaves agroinfiltrated with *Agrobacterium* mixture harboring BYSMV-agMR combinations containing an empty vector (EV), HvCCR4, HvCCR4$^N$, HvCCR4$^C$, or HvCCR4$^{mEEP}$. Representative images were taken at five dpi. Bar = 1 mm. (**C**) Western blotting analysis showing accumulation of RFP, N, P, and HvCCR4 proteins in the leaves shown in panel (**B**) with rabbit α-RFP, α-N, α-P, or α-Flag protein antibodies, respectively. *N. benthamiana* leaves infiltrated with the pGD vector were mock controls. (**D**) qRT-PCR analysis of minigenome RNA replication in the samples shown in panel (**B**). (**E**) qRT-PCR analysis of normalized levels of RFP transcription by comparing the relative levels of mRNA versus minigenome RNA in the samples shown in panel (**B**). EF1A served as an internal control gene. The values of viral replication and transcription in leaf samples agroinfiltrated with the EV plasmid were set to 1. Error bars indicate standard errors of three independent experiments. Letters above the bars indicate statistical significance (p<0.01) evaluated by Turkey's Multiple Comparison Test analysis.

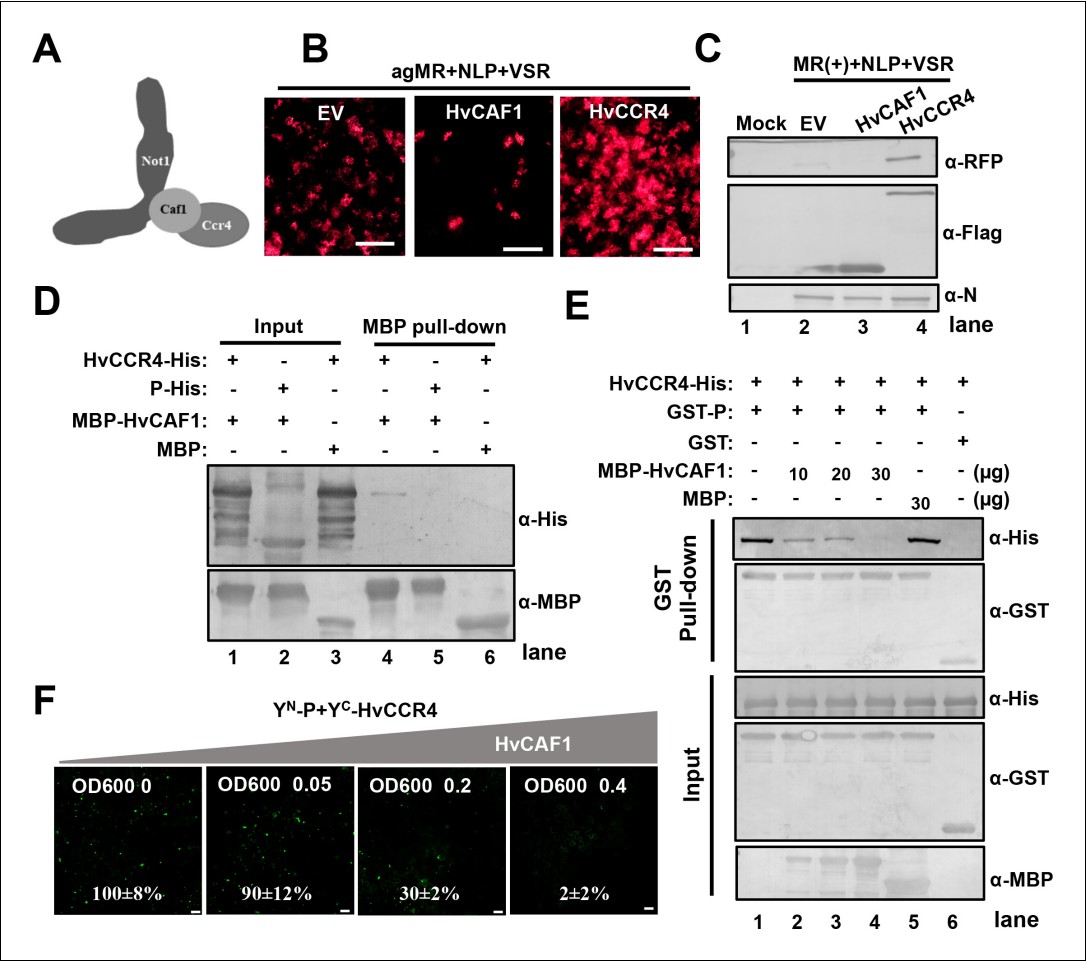

**Figure 5.** HvCAF1 negatively regulates minigenome RNA replication. (**A**) Schematic representation of CAF1 linking CCR4 and the Not1 scaffold in the CCR4-NOT complex. (**B**) RFP foci in *N. benthamiana* leaves agroinfiltrated with *Agrobacterium* mixture harboring BYSMV-agMR combinations or the empty vector (EV), HvCAF1, or HvCCR4. Representative images were taken at five dpi. Bar = 1 mm. (**C**) Western blotting analysis showing accumulation of RFP, HvCAF1, and HvCCR4 proteins in the leaves shown in panel (**B**) with rabbit antibodies against RFP and Flag proteins, respectively. (**D**) MBP pull-down assays showing the in vitro interaction of HvCAF1 with HvCCR4, but not with BYSMV P. CCR4-His or P-His were incubated with MBP-HvCAF1 or MBP with anti-MBP beads. Pull down or input proteins were detected by western blotting with anti-His or anti-MBP antibodies. (**E**) Competitive GST pull-down assays in vitro. GST-P and HvCCR4-His were pull-downed with glutathione-Sepharose beads incubated with increasing concentrations (10, 20, and 30 µg) of MBP-HvCAF1 or free MBP (30 µg). Input and GST pull down proteins were analyzed with anti-His, -GST, or -MBP antibodies. Free GST and HvCCR4-His served as negative controls. (**F**) Competitive BiFC assays in vivo. *N. benthamiana* leaves were infiltrated with *Agrobacterium* strains expressing $Y^N$-P and $Y^C$-HvCCR4 with increasing concentrations (OD600 0, 0.05, 0.2, 0.4) of *Agrobacterium* expressing HvCAF1. The bottom values represent relative granule numbers of different treatments. Values in leaf samples agroinfiltrated with HvCAF1 (OD600 0) were set to 100%. Error bars indicate standard errors of three independent experiments. Images were taken at three dpi by a Leica laser scanning microscope. Bar = 50 µm.

The online version of this article includes the following figure supplement(s) for figure 5:

**Figure supplement 1.** Phylogenetic tree (**A**) and sequence alignments (**B**) of plant CAF1 orthologues.
**Figure supplement 2.** BiFC assays examining interactions between the P, HvCCR4 and HvCAF1 proteins.
**Figure supplement 3.** Western blotting analysis of accumulation of expressed proteins in competitive BiFC assays.
**Figure supplement 4.** HvCAF1 negatively affect viroplasm-like structure formation.

## Overexpression of HvCCR4 facilitates full-length BYSMV infection in barley plants

Recently, we rescued BYSMV infections from the full-length cDNA clones and developed a versatile expression platform for functional studies of foreign proteins in barley plants and SBPHs (*Gao et al., 2019*). In the current study, we have used the BYSMV vector to overexpress HvCCR4 to investigate HvCCR4 functions in authentic virus infections. To this end, we generated pBYR-HvCCR4 by replacing the GUS ORF of pBY-GUS with the HvCCR4 ORF (*Figure 6A*). In addition, these vectors also contain an RFP insertion to monitor virus infections in barley plants (*Figure 6A*).

To rescue the full length BYSMV virus, the pBYR-HvCCR4 or pBYR-GUS plasmids were co-expressed with the pGD-NLP and pGD-VSRs plasmids in *N. benthamiana* leaves by agroinfiltration. At 12 dpi, RFP fluorescence foci in BYR-HvCCR4-infected *N. benthamiana* tissue were more numerous than tissue infiltrated with BYR-GUS (*Figure 6B*). Western blotting analyses consistently indicated that BYR-HvCCR4 infiltrations had a 110% increase of RFP accumulation in infected leaves than those infected with BYR-GUS (*Figure 6C*). We further expressed HvCCR4$^{mEEP}$ in the BYSMV vector, revealing that BYR-HvCCR4$^{mEEP}$ exhibited obviously reduced BYSMV infections than BYR-HvCCR4 (*Figure 6—figure supplement 1*), which is consistent with the agMR assays as shown in *Figure 4*. Collectively, these results suggest that HvCCR4 facilitates BYSMV infections in *N. benthamiana* leaves.

For transmission of BYSMV to barley plants, crude extracts of *N. benthamiana* leaves infected with BYR-HvCCR4 or BYR-GUS were injected into healthy SBPH thoraxes as described previously (*Gao et al., 2019*). After a 10 day incubation period in healthy rice plants, SBPHs were transferred to healthy barley plants for a two-day inoculation period. At 15 dpi, newly emerging leaves of BYR-HvCCR4-infected plants developed more severe symptoms and exhibited more intense RFP fluorescence than BYR-GUS-infected plants (*Figure 6D and E*). Western blotting analysis show that accumulation of RFP and BYSMV N proteins increased by 133% and 75% in newly emerging leaves of BYR-HvCCR4-infected barley plants compared with those of BYR-GUS-infected plants (*Figure 6F*). In addition, accumulation of HvCCR4 was higher in BYR-HvCCR4-infected barley plants than in BYR-GUS-infected plants (*Figure 6F*). These results clearly show that overexpression of HvCCR4 facilitates the virus infections in barley plants.

## CCR4 knockdown inhibits BYSMV infection of insect vectors

BYSMV is a cross-kingdom virus infecting both plants and insects. Therefore, we examined the effects of CCR4 on BYSMV infection of insect vectors. To monitor virus infections in SBPHs, we used a recombinant BYSMV vector with an RFP insertion (BY-RFP) that has been described previously (*Figure 7A*; *Gao et al., 2019*). We first cloned the SBPH CCR4 ORF (LsCCR4) that contains the N-terminal LRR region and the C-terminal EEP domain (*Figure 1—figure supplements 1* and *2*). In addition, the LsCCR4 EEP domain contains the conserved Asn260 and Glu305 residues that are required for deadenylase activity (*Figure 1—figure supplement 2C*). GST pull-down analysis showed that GST-LsCCR4 was pull-downed with P-His, rather than the control but not the control GST and GFP-His (*Figure 7B*). BiFC assays consistently showed that LsCCR4 and its N terminal LRR domain associated with BYSMV P into some cytoplasmic punctuates (*Figure 7—figure supplement 1*). Moreover, the NCMV P protein interacts with LsCCR4 in BiFC assays (*Figure 1—figure supplement 7*).

Next, we *LsCCR4* dsRNA to evaluate knocked down LsCCR4 functions during BY-RFP insect infections. For these experiments, crude extraction of BY-RFP-infected barley leaves were mixed with synthesized *GFP* or *LsCCR4* dsRNAs (final concentrations, 2 μg/μL), and coinjected with 13.8 nl of the mixtures extractions into healthy SBPH thoraxes. Then, the injected insects were maintained on healthy rice seedlings and RFP fluorescence was monitored at 3-, 7-, and 10- dpi (*Figure 7C*). RFP fluorescence of leafhoppers microinjected with dsGFP was first observed at three dpi and became disseminated throughout infected SBPHs at 7- and 10- dpi (*Figure 7C*). In contrast, very faint RFP fluorescence was observed in the SBPHs with dsCCR4 microinjection even at 10 dpi (*Figure 7C*). Western blotting analysis consistently showed that microinjection with dsCCR4 efficiently inhibited accumulation of BYSMV N and RFP by at least 36% compared with those of dsGFP treatment (*Figure 7D*). QRT-PCR analysis show that microinjection with dsCCR4 resulted in reduction of N, RFP and CCR4 mRNA levels to approximately 30%, 60%, and 20% of dsGFP treatment, respectively

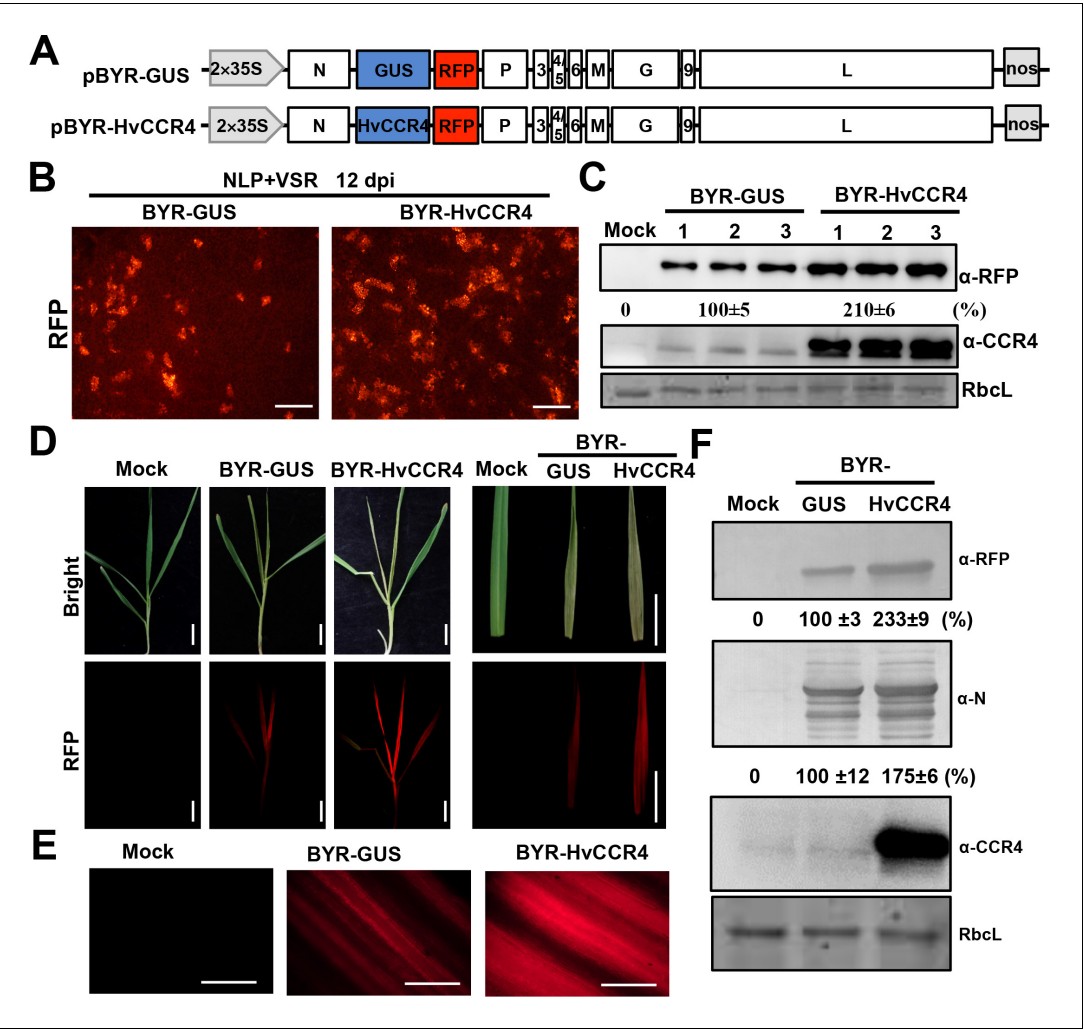

**Figure 6.** BYSMV-mediated HvCCR4 overexpression enhances virus pathogenesis in barley plants. (**A**) Schematic diagrams of BYSMV pBYR-GUS and pBYR-HvCCR4 derivatives harboring GUS and HvCCR4 ORFs. (**B**) RFP foci in *N. benthamiana* leaves infiltrated with *Agrobacterium* strains containing pGD-NLP, pGD-VSRs, pBYR-GUS or pBYR-HvCCR4 plasmids. RFP fluorescence was photographed after 12 dpi with a fluorescence microscope. Bar = 1 mm. (**C**) Western blotting analysis showing accumulation of RFP and CCR4 with anti-RFP and -CCR4 antibodies. Relative accumulation of RFP from three repetitions are shown at the bottom of RFP panels. (**D**) Disease symptoms and RFP fluorescence of barley plants infected with BYR-GUS or BYR-HvCCR4 at 15 dpi. Bar = 1 cm. (**E**) RFP fluorescence of systemically infected barley leaves with BYR-GUS or BYR-HvCCR4 at 15 dpi. Bar = 1 mm. (**F**) Western blotting analysis showing accumulation of RFP, N, and HvCCR4 proteins in the leaves shown in panel D. The mean relative values of three experiments were shown under the results. The mean values in BYR-GUS infected samples were set as 100%.

The online version of this article includes the following figure supplement(s) for figure 6:

**Figure supplement 1.** HvCCR4$^{mEEP}$ overexpression did not enhances virus pathogenesis in *N. benthamiana* leaves.

(*Figure 7E*). These results clearly demonstrate that knockdown of *LsCCR4* inhibits BYSMV infections in insect vectors.

After a 10 day incubation period, the BY-RFP-infected SBPHs were transferred to healthy barley plants for a 2 day inoculation period. At 15 dpi, plants inoculated by dsGFP-treated insects exhibited classical yellow striate symptoms and RFP fluorescence (*Figure 7F and G*). In contrast, symptoms and RFP fluorescence were significantly reduced in plants inoculated by dsCCR4-treated insects (*Figure 7F and G*). Western blotting analysis consistently showed that dsCCR4 treatment efficiently

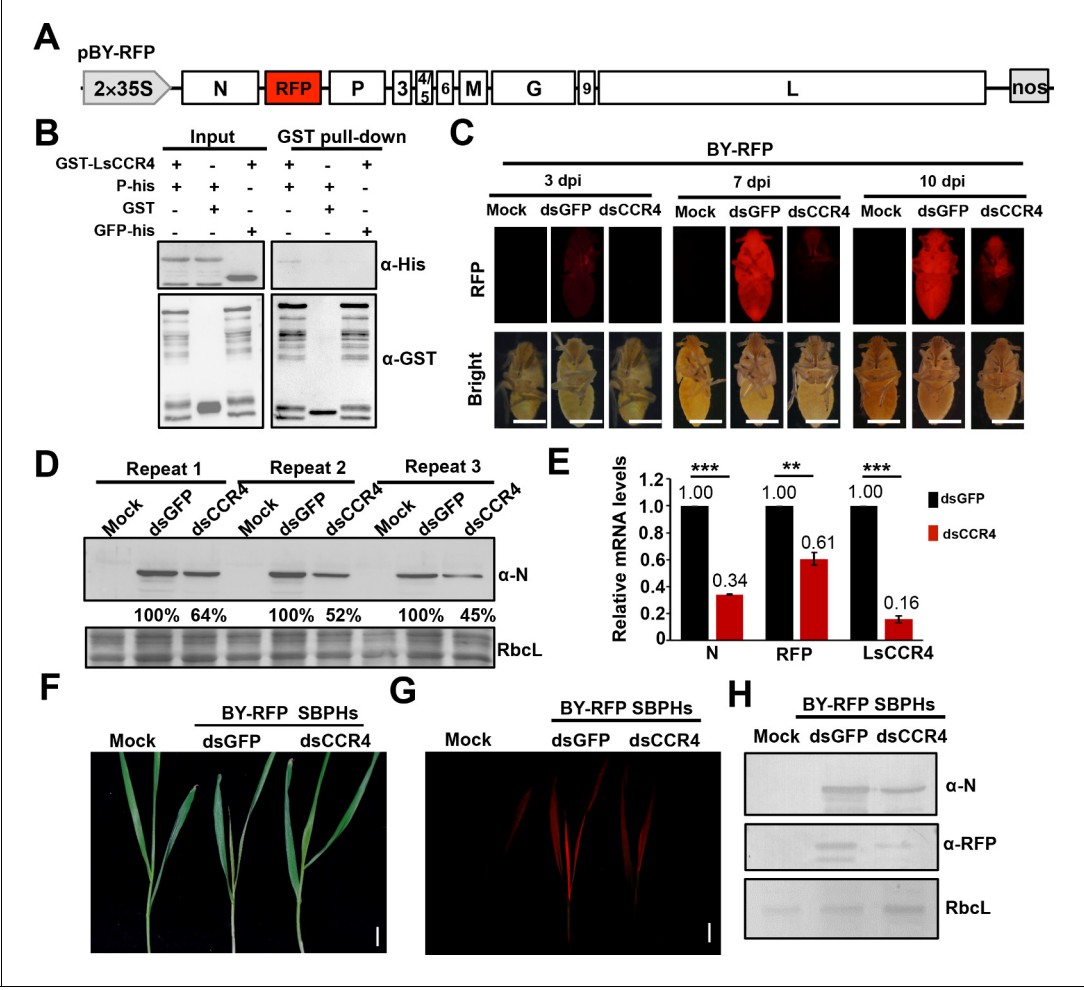

**Figure 7.** Requirement of the planthopper LsCCR4 protein for efficient BYSMV infection in SBPHs.  (**A**) Schematic diagrams of the BYSMV pBY-RFP derivative containing an RFP gene between the N and P genes of the antigenome cDNA. (**B**) GST pull-down analysis of interactions between BYSMV P and LsCCR4. GST and GFP-His served as negative controls. (**C**) RFP fluorescence of SBPHs microinjected with crude extracts of BY-RFP-infected barley leaves and mock buffer, dsGFP, or dsCCR4. At 3-, 7-, and 10- dpi, the insects were photographed with a fluorescence microscope. Representative confocal images from 30 insects are shown. Bar = 1 mm. (**D**) Western blotting analysis of BYSMV N protein accumulation in the samples shown at 10 dpi in panel (**C**) with the anti-N polyclonal antibody. Protein accumulation in the dsGFP-injected samples was set to 100%. Three independent repeats from 30 insects were shown. (**E**) qRT-PCR analysis of N, RFP and CCR4 mRNA accumulation in the samples shown in panel (**C**) at 10 dpi. EF1A acted as an internal control gene and the values in the dsGFP-injected insects were set to 1. Error bars indicate the standard error of the mean values of three independent experiments. Statistical significance was determined by Student's t test (** p-value<0.01; *** p-value<0.001). (**F**) Symptoms in systemically infected barley leaves at 15 days after feeding with BY-RFP infected SBPHs treated with dsGFP or dsCCR4. Bar = 2.0 cm. (**G**) RFP fluorescence in the barley leaves shown in panel (**F**). Bar = 2.0 cm. (**H**) Western blotting analysis of accumulation of BYSMV N and RFP in the samples shown in panel (**F**). Rubisco complex large subunit (RbcL) was detected with Stain-Free technology as equal loading controls.

The online version of this article includes the following figure supplement(s) for figure 7:

**Figure supplement 1.** BiFC assays examining interactions between BYSMV P and LsCCR4.

inhibited accumulation of BYSMV N and RFP in barley plants (*Figure 7H*). These results therefore suggest that LsCCR4 is important for BYSMV infection of planthoppers and transmission to uninfected barley.

## CCR4 triggers decay of the N-bound cellular mRNA

Previous studies have shown that CCR4 does not have RNA-binding activities and requires some RNA-binding proteins as accessory proteins to degrade specific mRNAs (*Yamaji et al., 2017*; *Zhu et al., 2018*; *Arae et al., 2019*; *Meijer et al., 2019*; *Webster et al., 2019*). Therefore, we

examined whether N, P, and HvCCR4 have unspecific RNA-binding activities using Northwestern blotting assays using 6 × His tagged GFP, N, P, and HvCCR4 purified from *E. coli*. The same amounts (5 µg) of 6 × His tagged GFP, N, P, and HvCCR4 were used for Northwestern blotting with digoxigenin-labeled luciferase (Luc) mRNA. As expected, the BYSMV N protein exhibited high RNA-binding affinity for Luc mRNA (*Figure 8A*). However, the GFP, P, or HvCCR4 proteins failed to bind Luc mRNA (*Figure 8A*). These results indicate that HvCCR4 has no RNA-binding activity as described previously (*Yamaji et al., 2017*; *Zhu et al., 2018*; *Arae et al., 2019*; *Meijer et al., 2019*; *Webster et al., 2019*), and that the BYSMV N protein can bind mRNAs nonspecifically in vitro.

To examine the deadenylation activity of HvCCR4 in vitro, a 5'-fluorescein isothiocyanate-labeled RNA (5'-UCUAAAUAAAAAAAAAAAAAAAAAAAAA-3') was commercially synthesized as a substrate for HvCCR4. The labelled RNA probe was not obviously degraded after 30 min of incubation with the N–P complex, whereas approximately 44% of substrate remaining was not degraded after 30 min of incubation with HvCCR4 alone (*Figure 8B and C*). In contrast, only 17% of substrate remaining was detected at 30 min after incubated with the N–P complex and HvCCR4 (*Figure 8B and C*), indicating that the N–P complex enhanced the deadenylation activity of HvCCR4. Moreover, CCR4$^{mEEP}$ containing alanine-substituted mutants in the conserved Asn260 and Glu305 residues did not exhibit deadenylase activity even in the presence of the N–P complex (*Figure 8D*). Thus, the N–P complex accelerates the HvCCR4-mediated RNA decay by facilitating HvCCR4 associations with RNA substrate.

Our results above show that the BYSMV P protein interacts with CCR4, a host decay machinery protein, for optimal virus replication. These findings prompted us to investigate whether BYSMV P recruits host CCR4 to the N$^0$–P protein complexes that specifically triggers turnover of the N-associated cellular mRNAs. It is noteworthy that gRNAs or agRNAs RNAs functioning in replication of BYSMV or minireplicons do not contain Poly(A) tails, and hence are not CCR4 substrates. Thus, CCR4-mediated turnover of the N-bound cellular mRNAs could potentially release RNA-free N$^0$ to specifically bind genome RNA. To test this hypothesis, Agrobacterium harboring plasmids for expression of agMR, N, P, HvCCR4 and HvCCR4$^m$ was mixed in different combinations and coinfiltrated into *N. benthamiana leaves*. At two dpi, Co-IP assays showed that the N-Flag protein could be co-precipitated with P, HvCCR4, and HvCCR4$^{eEEP}$ proteins (*Figure 8E*). Then, we isolated N-bound RNAs, followed by measuring the co-purified the relative level of agMR versus the EF1A mRNA by qRT-PCR analyses to assess the relative levels of copurified agMRs versus the EF1A mRNA. In comparison with the N protein expression alone, the P co-expression enhanced the ratios of the N binding agMR by 37%, whereas co-expression of P and HvCCR4 substantially increased the ratio by 133% (*Figure 8F*). In contrast, the HvCCR4$^m$ protein could not improve the specificity of the BYSMV N protein to genome RNAs (*Figure 8F*). These results demonstrate that CCR4 improves the binding specificity of the BYSMV N protein to genome RNAs.

During the rhabdovirus replication process, the P protein binds the RNA-free N$^0$ proteins in the N$^0$-P complex to facilitate N$^0$ encapsidation nascent gRNAs and agRNAs (*Ivanov et al., 2011*). The rhabdovirus replication process requires continuous production of soluble and RNA-free N$^0$ proteins (*Masters and Banerjee, 1988*; *Peluso and Moyer, 1988*). It has been shown that BYSMV P binds to RNA-free N$^0$ in the N$^0$–P complex to prevent the N polymerization and non-specific binding of the N protein to host cellular RNAs (*Mavrakis et al., 2006*; *Chen et al., 2007*; *Leyrat et al., 2011b*). Our results demonstrate that the BYSMV P protein tethers CCR4 and the BYSMV N protein to trigger turnover of N-bound cellular mRNAs, thereby releasing the N$^0$ protein to specifically encapsidate viral gRNA to facilitate virus replication (*Figure 8G*).

## Discussion

To date, plant CCR4 proteins have been identified as an important RNA decay factor regulating various plant development scenarios (*Suzuki et al., 2015*; *Arae et al., 2019*), and has been shown to have important roles in plant immunity (*Guo et al., 2018*; *Yu et al., 2019*). However, understanding of CCR4 functions in virus infections is limited. In the current study, we have demonstrated that plant HvCCR4, a major cytoplasmic deadenylase, was hijacked from P bodies into BYSMV viroplasm-like bodies by the BYSMV P protein (*Figures 1* and *2*). Overexpression of HvCCR4 enhanced replication of BYSMV minigenome and full-length virus in *N. benthamiana* leaves and barley plants. Moreover, the enhancement of BYSMV replication requires CCR4 deadenylase activity and this activity is

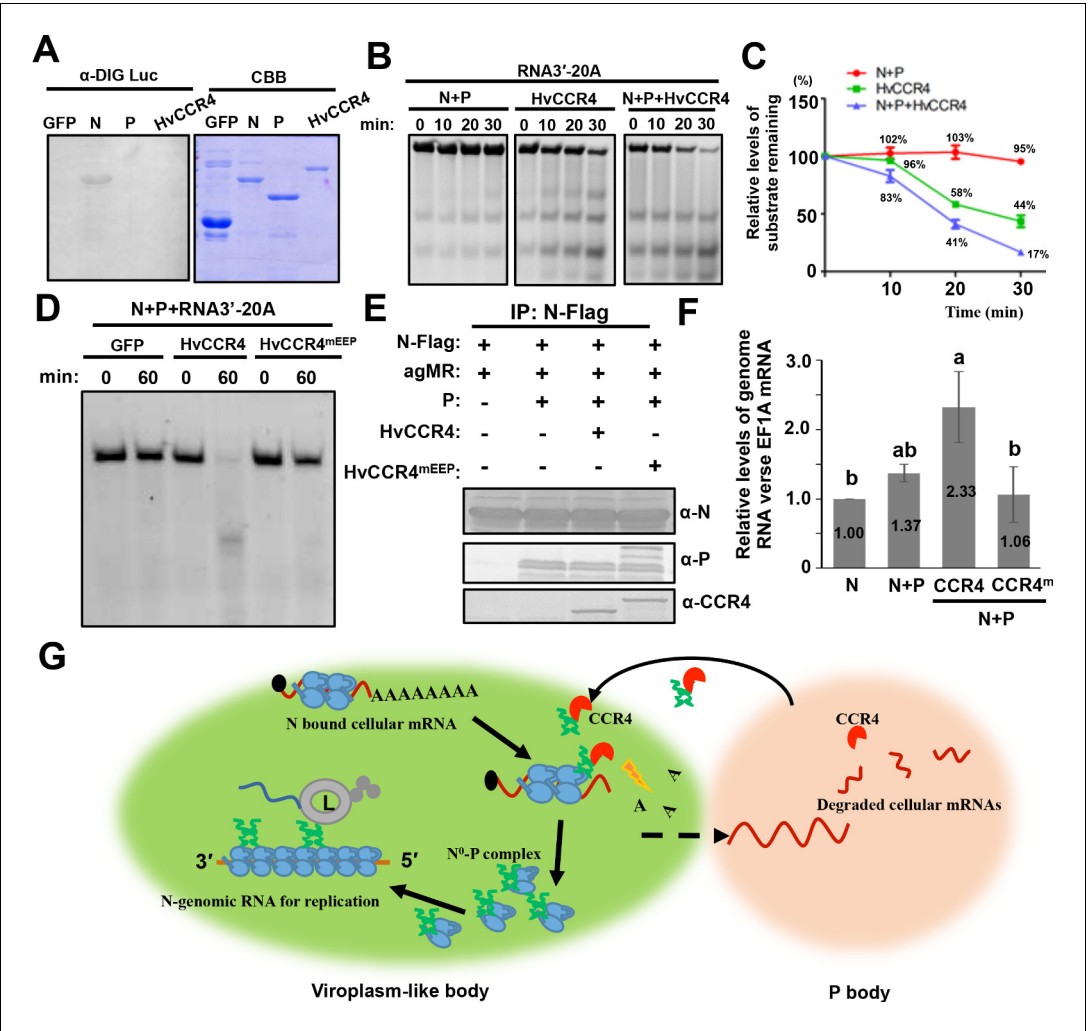

**Figure 8.** CCR4 facilitates binding of BYSMV N with viral genome RNAs by triggering N-bound cellular mRNA decay. (**A**) Northwestern blotting assays detecting non-specific RNA-binding abilities of HvCCR4, BYSMV N, P protein by digoxigenin-labeled Luciferase mRNA (See Materials and methods for details). (**B**) BYSMV N and P in vitro facilitation of deadenylase activity of purified HvCCR4 protein. The 5'-fluorescein isothiocyanate-labeled RNA substrate (RNA 3'−20A) was incubated with different protein combinations as indicated. At different time after incubation, the RNA substrate was analyzed by denaturing PAGE. Relative accumulation of undegraded RNA substrate was examined by Image J. The 0 min after incubation with proteins was set as one unit. (**C**) Degradation of the RNA 3'−20A substrate. The graphs plot the relative accumulation of remaining substrates estimated by electronic autoradiography in the gels of panel D. Data points represent mean values from three independent repeats. Error bars indicate Error bars indicate standard errors of three independent experiments. The curve diagrams were drawn with GraphPad Prism. (**D**) The N260 and E305 residues of HvCCR4 are required for the deadenylase activity of purified HvCCR4 protein in vitro. (**E**) and (**F**) HvCCR4 increases the binding specificity of the N protein to minigenome RNA. *N. benthamiana* leaves were infiltrated with *Agrobacterium* strains for expression of agMR, N-Flag, and other protein combinations as indicated. At three dpi, the N-Flag protein was immunoprecipitated from the infiltrated leaves with the anti-Flag M2 affinity gel. Western-blot analysis of N, P and CCR4 protein accumulation in the IP products (**D**). Analysis of N-Flag bound RNA from IP products by qRT-PCR to determine the ratios of N bound gRNA and host EF1A mRNA (**E**). Numbers above the graph show the mean values of three independent experiments. Statistical significance (p<0.05) was evaluated by two-way ANOVA with Tukey's test. (**G**) CCR4-mediated mRNA decay model for optimal BYSMV replication. Host CCR4 proteins hijacked by the BYSMV phosphoprotein trigger degradation of N-bound cellular mRNAs to release RNA-free N protein for enhanced encapsidation and replication of viral genomic RNA.

The online version of this article includes the following figure supplement(s) for figure 8:

**Figure supplement 1.** HvCCR4 did not exhibit obvious activities involved in RNAi.

abolished when the conserved Asn260 and Glu305 residues were replaced with alanines (*Figure 2*). BYSMV replication was inhibited when *LsCCR4* was knocked down in SBPH vectors (*Figure 7*). Here, these findings and those described below provide a novel strategy whereby BYSMV co-opts RNA decay machineries for virus infections.

PBs are mainly involved in RNA decay and microRNA mediated silencing which have been proposed as an antiviral strategies (*Lloyd, 2013*; *Reineke and Lloyd, 2013*). Genetic analysis has shown that RNA decay plays an important role in plant defense responses against virus infections by affecting viral mRNA stability (*Jaag and Nagy, 2009*; *Ma et al., 2015*; *Li and Wang, 2018*; *Garcia-Ruiz, 2019*). These studies mostly focus on the 5′ to 3′ RNA decay components, including exoribonuclease 4 (XRN4) and decapping protein 2 (DCP2) in limiting virus accumulation (*Jaag and Nagy, 2009*; *Ma et al., 2015*; *Li and Wang, 2018*). Correspondingly, some virus proteins, like potyviral HC-Pro and genome-linked protein (VPg), can interfere with RNA decay through interactions with XRN4 and DCP2 (*Li and Wang, 2018*). Compared to accumulating genetic evidence for antiviral RNA decay activities, less is known about the pro-viral activities of RNA decay in plant virus infections. In the case of BYSMV, we show that the BYSMV P protein interacts directly with host CCR4 to recruit CCR4 from PBs to BYSMV replication bodies containing N, L, P, and CCR4, are adjacent to P bodies marked by DCP1-mCherry. The results also suggest that the P protein recruits the protein from PBs to BYSMV replication bodies, indicating that cellular P bodies have important roles in viral replication. For example, BYSMV N-bound cellular mRNAs were first subjected to CCR4-mediated deadenylation, and then transferred to adjacent P bodies for RNA decay. Alternatively, the CCR4 protein may tether the BYSMV replication bodies and cellular P bodies to facilitate transfer of N-bound cellular RNAs to P bodies for RNA decay. In addition, RNA decay has shown to be related with RNA silencing in previous studies (*Christie et al., 2011*; *Li and Wang, 2018*; *Moreno et al., 2013*). Thus, CCR4 in the decay bodies might be a suppressor of RNA silencing to interfere antiviral RNA silencing, thereby promoting viral replication. In this study, we found that GFP-P bodies are adjacent to but not overlapped with SGS3-RFP-labelled siRNA bodies (*Figure 8—figure supplement 1*). However, we used co-infiltration assays to reveal that neither dsRNA-induced CCR4 silencing or CCR4 overexpression has suppressor activities of GFP-induced silencing as the TBSV P19 suppressor (*Figure 8—figure supplement 1B and C*). Nonetheless, we will examine these possibilities in future studies.

In the CCR4–NOT1 complex, both CCR4 and CAF1 are functional deadenylases that belong to the EEP and DEDD (Asp-Glu-Asp-Asp) families, respectively (*Dlakić, 2000*; *Zuo and Deutscher, 2001*; *Niinuma et al., 2016*). HvCAF1 does not interact with BYSMV P and plays a negative role in the BYSMV replication (*Figure 5*), and hence is functionally distinct from the CCR4–P interactions and proviral function. One interpretation of the negative role of HvCAF1 is that HvCAF1 sequesters the HvCCR4 protein from the BYSMV P protein, thus interfering with the proviral function of HvCCR4 in BYSMV replication. Moreover, these results suggest that BYSMV P selectively hijacks the CCR4 protein, but not the complete CCR4–NOT1 complex to facilitate optimal virus replication. In insects, LsCCR4 contains a conserved LRR domain present in yeast and human homologs that mediates interactions with the BYSMV P protein (*Figure 7*). Furthermore, LsCCR4 is required for BYSMV replication in insects (*Figure 7*). Thus, these results suggest that the BYSMV P protein can interact with the two distinct N-terminal motifs of plant and insect CCR4s, which may be an important for transkingdom BYSMV infection of plant hosts and insect vectors.

As obligate organisms, viruses encode limited functional proteins, and must usurp host cellular resources for replicative advantages. Thus, identifying host factors involved in viral replication process is critical for understanding molecular mechanisms of viral pathogenesis (*Wang, 2015*; *Hashimoto et al., 2016*). Rhabdovirus replication requires viral gRNA and agRNA and three core virus proteins, consisting of the N, P, and L proteins. The viral RNA encapsidated by the N protein serves as a template for virus transcription and replication. However, rhabdovirus N proteins have a strong unspecific affinity for cellular RNA (*Figure 8A*). Thus, for optimal viral replication, N proteins must be prevented from binding cellular RNAs. Previous studies about animal rhabdoviruses have demonstrated that rhabdovirus P binds to the nascent $N^0$ molecule to form soluble $N^0$–P complexes that maintains the $N^0$ molecule in an RNA-free state (*Masters and Banerjee, 1988*; *Peluso and Moyer, 1988*; *Mavrakis et al., 2006*; *Leyrat et al., 2011b*). However, it remains unknown whether N bound cellular RNA is degraded before interactions between the P and $N^0$ proteins. Our recent work has shown that BYSMV P forms trafficking bodies on the ER/actin networks to recruit BYSMV L,

$N^0$ or N/cellular RNA during viral RNA synthesis (*Fang et al., 2019*). The trafficking P bodies recruit host CCR4 (*Video 4*), indicating that CCR4 is a host factor required for viral replication. In BYSMV P bodies, CCR4 deadenylase activities could remove poly (A) tails to initiate turnover of N-bound mRNA, and free $N^0$ protein for P protein associations (*Figure 8G*). In agreement with this proposed model, we have shown that HvCCR4 can prevent N binding to cellular mRNAs through its deadenylation function. Thereafter, the released RNA-free $N^0$ to bind the P protein and specifically encapsidate viral genomic and antigenomic RNAs to provide functional templates for optimal replication (*Figure 8G*).

In conclusion, we have demonstrated for the first time that a rhabdovirus P protein can co-opt host CCR4 proteins for replication in plant and insect hosts. Host CCR4 protein mediated deadenylation is utilized by rhabdoviruses to prevent the N protein from binding to cellular mRNAs and to maintain specific encapsidation of viral genomic and antigenomic RNAs with the N protein to provide replication templates. In future studies, we plan to examine whether BYSMV P–CCR4 interactions interfere with cellular PB assembly and functions.

## Materials and methods

### Plant materials and virus inoculation by insect transmission

*N.N. benthamiana* and barley (Golden promise) plants were grown on soil at 25 ± 2℃ with a day/night cycle of a 14/10 hr. SBPHs were isolated from Hebei province, China and reared in illumination incubators. BYSMV was initially isolated from wheat fields in Hebei province, China and maintained in barley plants as previously described (*Di et al., 2014*; *Yan et al., 2015*). SBPH transmission of BYSMV by SBPHs was performed as described previously (*Cao et al., 2018*; *Gao et al., 2019*). Briefly, second-instar nymphs of SBPHs were collected, anesthetized on ice, and microinjected with crude extracts of infected leaves (13.8 nL per insect) with a Nanoinject II auto-nanoliter injector (Drummond Scientific Company, Broomall, PA, USA). After a 10 day incubation period on uninfected rice seedlings, the injected nymphs were transferred to healthy barley plants for 2 day inoculation access period. Observation of symptom expression and virus analysis were conducted at about 15 dpi.

### Plasmid constructions

*HvCCR4* (AK374808), *HvCAF1* (AK361706), *LsCCR4* (RZF46990.1) and BYSMV (KM213865) GenBank accession sequences were used throughout this study. To clone *HvCCR4*, *HvCAF1,* and *LstCCR4*, total RNA was extracted from barley leaves or insects, and used as templates for reverse transcription PCR (RT-PCR) with the specific primers shown in *Supplementary file 2*. The full length *HvCCR4* ORF, the *HvCCR4* N terminus (*HvCCR4$^N$*) and C terminus (*HvCCR4$^C$*), as well as the *HvCCR4* point mutant (N260A/E305A) were cloned into pMDC32−3 × Flag (*Wang et al., 2018*), modified from pMDC32 (*Curtis and Grossniklaus, 2003*). For biomolecular fluorescence complementation (BiFC) assays, the full length *HvCCR4* ORF, *HvCCR4$^N$* and *HvCCR4$^C$* sequences were introduced into pSPYNE or pSPYCE vectors (*Walter et al., 2004*). For subcellular localization assays, the full length *HvCCR4* ORF was engineered into pSuper1300-mCherry (*Jin et al., 2018*) and pGDGm (*Goodin et al., 2002*). For protein purification, the BYSMV P ORF was cloned into pGEX-KG to express GST-P protein in *E. coli*, and negative controls used vector encoding GST. The full length *HvCCR4* ORF was recombined into the pET30a vector to express the 6 × His tagged fusion proteins. All primers used to construct these vectors are listed in *Supplementary file 2*.

### Confocal laser scanning microscopy

Subcellular localization and BiFC images were captured at 2 days post agroinfiltration with a Leica TCS-SP8 confocal laser scanning microscope as described previously (*Fang et al., 2019*). CFP, GFP, YFP, and mCherry were visualized at excitations of 440 nm, 488 nm, 514 nm, and 543 nm, respectively. Videos were obtained from 50 frames (3 s/frame) using time series programs in the Leica TCS-SP8. ImageJ software was used to edit videos playing seven frames per second.

## Western blotting analysis

Total proteins were extracted from agroinfiltrated *N. benthamiana* leaves, infected insects, or diseased barley plants in SDS buffer [10% β-mercaptoethanol, 100 mM Tris (pH 6.8), 20% glycerol, 4% SDS, and 0.2% bromophenol blue], separated in SDS-PAGE gels and transferred to nitrocellulose membranes. Protein accumulation was detected with corresponding anti-BYSMV N (1:3000), P (1:3000), RFP (1:2000), or HA (1:3000) polyclonal antibodies, and goat anti-rabbit IgG horseradish peroxidase conjugate (1:30000) was used as secondary antibodies, followed by incubation with Pierce ECL Plus chemiluminescent substrate before exposure to x-ray films.

## Co-IP assays

Co-IP assays were performed as described previously (*Zhang et al., 2017*). Briefly, *N.benthamiana* leaves were agroinfiltrated with expression vectors of CCR4-Flag with P-GFP, P6-GFP, or GFP. Agroinfiltrated leaves were homogenized in liquid nitrogen and extracted with IP buffer (25 mM Tris-HCl, pH 7.5, 1 mM EDTA, 150 mM NaCl, 1% Tween20, 10% [v/v] glycerol, 5 mM DTT, 2% (w/v) polyvinyl-polypyrrolidone (PVPP) and protease inhibitor cocktail). After centrifugation and filtration, the supernatants were mixed with anti-Flag M2 affinity gel (Sigma) at 4°C for 4 hr with gentle mixing, followed by centrifugation at 800 g for 1 min. The precipitate was washed three times with IP buffer and analyzed by western blotting analysis with anti-Flag or anti-GFP antibodies.

## GST pull down assays

GST pull down assays were performed as described previously (*Yang et al., 2018*). GST-P, HvCCR4-His, or LsCCR4-His fusion proteins were expressed and purified from *E. coli* strain BL21. GST or GST-P were incubated with HvCCR4-His or LsCCR4-His in 500 μL reaction buffer (50 mM Tris-HCl, pH7.5, 300 mM NaCl, 0.6% TritonX-100, 0.1% glycerol, 1 × cocktail) with glutathione-agarose beads at 4°C for 2 hr. After centrifugation at 800 g for 1 min, beads were washed five times with reaction buffer, boiled in SDS buffer for western blotting analysis with anti-GST (1:5000) and anti-His (1:5000) antibodies.

## Knockdown of *CCR4* and virus infections in *L. striatellus*

Virus crude extraction from infected plants, nymphs injection, as well as SBPH-mediated virus transmission have been described previously (*Gao et al., 2019*). To knock down *CCR4* in insects, *LsCCR4* and *GFP* control fragments fused with the T7 promoter at their two termini were amplified, and the resulting PCR products served as templates for in vitro synthesis of double-stranded RNA of *CCR4* (dsCCR4) and dsGFP using the T7 RiboMAX Express RNAi System kit (Promega). Virus extract of BY-RFP-infected barley leaves (100 mM Tris-HCl pH 8.4, 10 mM Mg acetate, 1 mM MnCl$_2$, 40 mM Na$_2$SO$_3$) were mixed with synthesized GFP or CCR4 dsRNAs (final concentration, 1.5 μg/μL), and 13.8 nl of the mixed extracts were microinjected into second instar nymph thoraxes as described previously (*Gao et al., 2019*). At 3-, 7-, and 10 days after microinjection, planthoppers were monitored with an Olympus FV1000 microscope, and then collected for viral proteins and RNA analysis.

## Real-time quantitative RT-PCR (qRT-PCR)

Total RNA was isolated from barley, insect, or MR-infiltrated leaves and treated with DNase I (Takara, China) for RT reactions using M-MLV Reverse Transcriptase (Promega, USA). The full-length MR RNA abundance was analyzed with BYS-RT-1 as RT primer and the PCR primer corresponding to the trailer fragment. Accumulation of the BYSMV N, RFP, and the EF1A mRNA were analyzed with oligo dT as RT primer and their specific primers. QPCR were carried out using 2 × SsoFast EvaGreen Supermix (Bio-Rad). The primers used in this study are listed in *Supplementary file 2*. At least three independent biological replicates were collected for biological statistics analysis.

## Deadenylation assays

In vitro deadenylase assays were performed as described (*Chou et al., 2017*). Briefly, commercial 5'-fluorescein isothiocyanate-labeled RNAs (final concentration, 0.5 μM; 5'-UCUAAA UAAAAAAAAAAAAAAAAAAA-3') served as a substrate for deadenylation assays. The labeled RNA substrate was incubated with HvCCR4-His or its derivative proteins in reaction buffer (50 mM Tris–HCl, pH 7.5, 50 mM NaCl, 1 mM MgCl2, 10% glycerol) at 37°C for different times as indicated.

Then, reaction mixtures were fractioned on denaturing polyacrylamide gels (16%, 7 M urea), and the RNA gels were analyzed with a fluorescence imager (Bio-Rad). To quantify the rates of deadenylated RNA, we fist quantified the 5'FAM-RNA band intensities by ImageJ software. The graphs plot the relative accumulation of remaining substrates estimated by electronic autoradiography in the gels. The values of 5'FAM-RNA band intensities at 0 min were set to 100%. Data points represent mean values from three independent repeats. The curve diagrams was drawn with GraphPad Prism.

## Northwestern blotting assays

Northwestern blotting assays were performed as described previously (*Zhang et al., 2017*). Briefly, 5 µg of GFP, N, or HvCCR4 proteins were separated in 12.5% SDS-PAGE gels and transferred to nitrocellulose membranes. The membranes were incubated in renaturation buffer (10 mM Tris-HCl, pH 7.5, 50 mM NaCl, 1 mM EDTA, 0.02% BSA, 0.02% PVP40, 0.02% Ficoll 4000, 0.1% Triton X-100) overnight at 4°C and a digoxigenin-11-UTP-labelled (Roche) Luc RNA probe was added into the reactions and maintained at 25°C for 3 hr. After washing three times, the membranes were blotted with the anti-digoxigen conjugated alkaline phosphatase (1:30000) and analyzed in an NBT/BCIP solution.

## In vivo N-bound RNA analysis

The BYSMV N-Flag and P proteins, CCR4, and minigenome RNA in different combinations were transiently expressed in agroinfiltrated *N.benthamiana* leaves. At three dpi, total proteins were extracted from the agroinfitrated *N.benthamiana* leaves and incubated with anti-Flag M2 affinity gel (Sigma) in IP buffer (25 mM Tris-HCl, pH 7.5, 1 mM EDTA, 150 mM NaCl, 1% Tween-20, 10% glycerol, 5 mM DTT, 2% polyvinylpolypyrrolidone and protease inhibitor cocktail) at 4°C for 4 hr. After washing five times, the IP products were used for protein detection by western blotting analysis with corresponding antibodies. In addition, IP product RNA was extracted with TRIzol Reagent for qRT-PCR analysis of minigenomic RNA and the *EF1A gene* accumulation with specific primers as shown in *Supplementary file 2*.

# Acknowledgements

We thank our colleagues Jialin Yu and Yongliang Zhang at China Agricultural University and Prof. Andrew O Jackson (Department of Plant and Microbial Biology, University of California, Berkeley, USA) for their helpful suggestions and constructive criticism. We thank Prof. Zhen Li for his support in LC-MS/MS analysis at the Mass Spectrometry Facility of China Agricultural University. This work was supported by Natural Science Foundation of China (31872920 and 31571978) to XBW, and National Science and Technology Major Project of the Ministry of Science and Technology of China (2016Z × 08002001) to CGH.

# Additional information

## Funding

| Funder | Grant reference number | Author |
| --- | --- | --- |
| National Natural Science Foundation of China | 31872920 and 31571978 | Xian-Bing Wang |
| National Natural Science Foundation of China | 31571978 | Xian-Bing Wang |

The funders had no role in study design, data collection and interpretation, or the decision to submit the work for publication.

## Author contributions

Zhen-Jia Zhang, Data curation, Formal analysis, Validation, Investigation, Visualization, Methodology, Writing - original draft, Writing - review and editing; Qiang Gao, Dong-Min Gao, Qing Cao, Yi-Zhou Yang, Investigation, Visualization, Methodology; Xiao-Dong Fang, Validation, Investigation, Visualization, Methodology; Zhi-Hang Ding, Investigation, Methodology; Wen-Ya Xu, Investigation,

Visualization; Ji-Hui Qiao, Validation, Investigation; Chenggui Han, Conceptualization, Funding acquisition, Validation; Ying Wang, Conceptualization, Validation, Writing - review and editing; Xuefeng Yuan, Visualization, Methodology; Dawei Li, Supervision, Validation, Project administration; Xian-Bing Wang, Conceptualization, Resources, Supervision, Funding acquisition, Validation, Writing - original draft, Project administration, Writing - review and editing

**Author ORCIDs**
Zhen-Jia Zhang (iD) https://orcid.org/0000-0003-0473-2738
Xian-Bing Wang (iD) https://orcid.org/0000-0003-3082-2462

**Decision letter and Author response**
Decision letter https://doi.org/10.7554/eLife.53753.sa1
Author response https://doi.org/10.7554/eLife.53753.sa2

## Additional files

### Supplementary files
• Supplementary file 1. List of the BYSMV P protein interacting barley proteins obtained in IP-MS assays.
• Supplementary file 2. Primers used in this study.
• Supplementary file 3. Key Resources Table.
• Transparent reporting form

### Data availability
All data generated or analysed during this study are included in the manuscript and supporting files.

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
