## [Decision Letter]

**Acceptance summary:**

This manuscript teaches us that a plant virus, BYMSV, uses its phosphoprotein (P) to hijack the host deadenylation factor CCR4 to promote viral replication. A particularly striking finding is that the virus uses the same trick to manipulate both its plant host and its insect vector. The reviewers were pleased that the authors were overall very aware of the many pitfalls inherent in the complex experimental setups and data analysis. The work advances our understanding of the molecular arms race in virus-host interactions and will be of great interest to the fields of viral interaction and RNA turnover.

**Decision letter after peer review:**

Thank you for submitting your article "A co-opted CCR4 RNA decay factor enhances plant cytorhabdovirus replication" for consideration by eLife. Your article has been reviewed by Detlef Weigel as Senior and Reviewing Editor and three reviewers. The following individuals involved in review of your submission have agreed to reveal their identity: Yi Li (Reviewer #3).

The reviewers have discussed the reviews with one another and the Reviewing Editor has drafted this decision to help you prepare a revised submission.

Summary:

RNA quality control and regulated RNA decay are conserved eukaryotic mechanisms that protect host cells from dysfunctional endogenous or invasive RNAs. Therefore, viruses have evolved strategies not only to evade, but also to manipulate these mechanisms in order to successfully complete their replication cycle.

In this manuscript, you address the roles of carbon catabolite repression 4 (CCR4) in a plant virus infection both in the host plant and the insect vector. In eukaryotes, mRNA deadenylation is primarily mediated by the CCR4-NOT complex in RNA processing bodies (PBs) known from yeast and human cells, but regulation of CCR4 and the role of its deadenylase activity during viral infections is not well understood. You discovered that BYMSV uses its phosphoprotein (P) to hijack host CCR4 to promote viral replication. You convincingly show that P protein interacts directly with CCR4 and recruits it into viroplasm-like bodies to improve viral RNA replication; P-recruited CCR4 proteins are responsible for turnover of the N-bound cellular mRNAs and thereby releasing RNA-free N protein to bind viral genomic RNA for optimal viral replication.

You present a potential mechanism for how P protein binds to RNA-free N protein (N^0^) to prevent binding to cellular mRNAs and to facilitate specific viral gRNA/agRNA interactions. A particularly nice aspect of the work is that you also tested CCR4 orthologues in the transmitting vector, which led to similar conclusions: CCR4 knockdown inhibits BYSMV infection of insects and significantly reduces virus transmissibility, indicating a potential evolutionarily conserved mechanism that BYSMV has adopted to outcompete and inhibit N protein binding of cellular mRNAs. You are overall very aware of the many pitfalls inherent in the complex experimental setups and data analysis. The manuscript is well written and does an especially effective job presenting complex interactions.

Overall, you present several interesting observations directly validated by combining approaches from molecular biology, cell biology and biochemistry. This allows you to propose a novel mechanism of how rhabodoviruses exploit the host RNA decay machinery for viral replication. The work advances our understanding of the molecular arms race in virus-host interactions and will be of great interest to the fields of viral interaction and RNA turnover.

While the reviewers were overall very positive, we ask you to conduct a few essential revisions.

Essential revisions:

1) You used transient expression to test N-bound RNA ratios in Figure 8E and F, with the key proteins all expressed from 35S promoters. To obtain more physiological relevant ratios, please repeat the experiments for Figure 8E and F using BYR-HvCCR4/HVCCR4^mEEP^-rescued barley plants.

2) Rhabdovirus P protein has been shown to form dimers through the central dimerization domain and the dimerization is important in the process of chaperoning nascent RNA-free N (N^0^) by forming the N^0^-P complex that prevents N^0^ from binding non-specifically to cellular RNAs, and attaches L protein polymerase complexes to the N-RNA complex. In Figure 1, you show that P interacts with CCR4 through the central and/or the C-terminal domain. Does the CCR4-P interaction interfere with the dimerization of P protein? Also, what is the effect on the function of P protein? Is the functional P form a dimer or a monomer?

3) The deadenylation assays showed strong and direct evidence that P protein could tether CCR4 and that the N protein triggers degradation of RNA containing poly(A). However, the quality of Figure 8B should be improved; no significant increase of the degradation products can be seen in the figure at 15, 30, and 60 minutes. You should also describe the method and details how to quantify the deadenylated RNA in Figure 8B and C.

4) RQC and RNA decay act as antiviral mechanisms, but they also play a role as endogenous suppressors of PTGS. This entails that HvCCR4 overexpression may promote viral replication through repressing PTGS. HvCCR4 is localized to viroplasm-like structures where viral mRNA is transcribed. In another scenario, it is possible that HvCCR4 degrades aberrant or excessive viral mRNAs to suppress antiviral RNA silencing, thereby promoting viral replication. *Arabidopsis* CCR4a is localized to both P-bodies and siRNA bodies (Moreno et al., 2013) and BYSMV infection promotes the generation of virus-derived siRNAs as shown in one of the authors' previous study. Do BYSMV P protein (or viroplasm-like structures) co-localize with siRNA bodies and HvCCR4? Is the accumulation of virus-derived siRNAs affected by HvCCR4 overexpression (Figure 6D and Figure 7F)? Please consider this possibility and discuss it in depth.

5) Figure 2D and E requires quantitative data showing how many foci were detected and how many were completely or adjacently co-localized in the absence/presence of P proteins.

[Editors' note: further revisions were suggested prior to acceptance, as described below.]

Thank you for resubmitting your work entitled "A co-opted CCR4 RNA decay factor enhances plant cytorhabdovirus replication" for further consideration by eLife. Your revised article has been evaluated by Detlef Weigel (Senior Editor) and a Reviewing Editor.

The manuscript has been improved but there are some remaining issues that need to be addressed before acceptance, as outlined by reviewer 4. Could you please address these in the final revision?

*Reviewer #1:*

Deadenylation of mRNA is the first and rate-limiting step of mRNA and is primarily mediated by the CCR4-NOT complex in RNA processing bodies (PBs). In this manuscript, Zhang et al. addressed the important roles of carbon catabolite repression 4 (CCR4) in a plant virus infection both in plant and insect vector. The quality of the manuscript is significantly improved after revisions. I recommend acceptance.

*Reviewer #2:*

The authors have successfully addressed most of my concerns, which improve the clarity and the significance of the study. Regarding the determination of the N-bound RNA ratio, I still think that in vivo date with stable lines would be much more convincing. However, in the revised manuscript, the additional explanations and experiments supporting the promotion of viral replication by HvCCR4 and the fact that the level of CCR4 does not affect RNA silencing strengthen the conclusion of this study revealing a new aspect of the function of a RNA decay component, HvCCR4, in the interaction between viruses and plants, and I agree with it.

*Reviewer #4:*

In this manuscript, Zhang et al. report that CCR4 RNA decay factor is hijacked by BYSMV P protein into viroplasm-like bodies to trigger degradation of cellular mRNAs bound by BYSMV N protein, allowing N protein to bind viral genomic RNA for viral replication. The findings deepen our understanding of the molecular mechanisms in host-virus arm race.

In general, the conclusions supported by the data they presented and the data are of good quality. The authors also addressed most of the concerns raised during the last round of review. Listed below are my comments:

1) The Title should be revised to reflect that CCR4 RNA decay factor is hijacked by BYSMV P.

2) CCR4 was identified to interact with BYSMV P protein through IP-MS, proteins identified in IP-MS should be listed in a table.

3) Figure1—figure supplement 6A, labeling for the fifth lane is missing. The bottom western blot image only has 4 lanes, other images have 5 lanes.

4) In Figure 6D-F, the authors show that barley plants infected with BYR-HvCCR4 developed more severe symptoms and have a higher level of BYSMV accumulation. Is it because that SBPH infected with BYR-HvCCR4 carries a higher amount of BYSMV?

5) For all the western blot results, the loading control should be specified. For some of the IP results, for instance, Figure 1C and D, only the symbol "+" was used. The symbol "-" was missing above the images.

The writing should be improved. The authors should check throughout the manuscript for writing problems. Here are some examples.

6) Introduction, "Thus, shortening of mRNA poly (A) tail..."

The word "thus" should be removed.

7) Introduction, "Thus, studying the interplay between.. . understanding the molecular mechanisms of host-pathogen interactions"

The sentence should be removed.

8) Subsection “BYSMV P interacts with barley HvCCR4 in vitroand in vivo*”*, "when N. benthamiana leaves... in contrast to the highly dynamic movement along ER tubules observed in the DMSO control"

The sentence should be rephrased.

9) Subsection “BYSMV P interacts with barley HvCCR4 in vitroand in vivo*”*, the results should be described at the end of the preceding paragraph.

10) Subsection “HvCAF1 has a negative effect on BYSMV minigenome replication”, "Western blotting analyses revealed that.. .in different expression level of HvCAF1-Flag"

The sentence should be rephrased.

11) Subsection “HvCAF1 has a negative effect on BYSMV minigenome replication”, "thereby weakening the HvCCR4-P interactions that compromise HvCCR4 proviral functions in BYSMV replication"

The sentence should be rephrased.

12) Subsection “CCR4 triggers decay of the N-bound cellular mRNA”, "exhibit specificity improvement”.

---

## [Author Response]

Essential revisions:1) You used transient expression to test N-bound RNA ratios in Figure 8E and F, with the key proteins all expressed from 35S promoters. To obtain more physiological relevant ratios, please repeat the experiments for Figure 8E and F using BYR-HvCCR4/HVCCR4^mEEP^-rescued barley plants.

Thanks for suggestions. As shown in Figure 4B, transiently expressed HvCCR4, but not HvCCR4^mEEP^, could significantly increase the replication of minireplicon genome. Therefore, different accumulation of agMR would affect the final ratios of N-bound agMR verse cellular RNA in the presence of HvCCR4 or HvCCR4^mEEP^. To rule out the different accumulation of viral genomic RNA, we did not express the BYSMV L protein in the Figure 8E and F. Therefore, accumulation of the agRNA in vivo is similar in different samples, because all agRNA is transcribed from 35S promoter without virus replication. Then, we carried out the IP assays using N-Flag to show HvCCR4, but not HvCCR4^mEEP^, could improve the binding specificity of N to agMR.

We also constructed the pBYR-HvCCR4^mEEP^ plasmid as the pBYR-HvCCR4 using the BYSMV vector. We first inoculated BYR-HvCCR4^mEEP^ and BYR-HvCCR4 in two half parts of *N. benthamiana* leaves to compare their infectivity. In consistence with transiently expressed HvCCR4^mEEP^ and HvCCR4 in minireplicon assays shown in Figure 4B, BYR-HvCCR4^mEEP^ exhibited a significantly reduced infectivity compared with BYR-HvCCR4, which was revealed by the RFP fluorescence and accumulation of RFP as shown in Figure 6—figure supplement 1. Since replication of BYR-HvCCR4^mEEP^ is very weak in *N. benthamiana* leaves, we cannot rescue the BYR-HvCCR4^mEEP^ infections in planthoppers and barley plants. We will examine the BYR-HvCCR4^mEEP^ infections in barley plants in future studies.

2) Rhabdovirus P protein has been shown to form dimers through the central dimerization domain and the dimerization is important in the process of chaperoning nascent RNA-free N (N^0^) by forming the N^0^-P complex that prevents N^0^ from binding non-specifically to cellular RNAs, and attaches L protein polymerase complexes to the N-RNA complex. In Figure 1, you show that P interacts with CCR4 through the central and/or the C-terminal domain. Does the CCR4-P interaction interfere with the dimerization of P protein? Also, what is the effect on the function of P protein? Is the functional P form a dimer or a monomer?

Thanks for detailed suggestions. To address this concern, we carried out competitive pull-down in vitro and BiFc assays in vivo, showing that overexpress CCR4 had no effect on self-interaction activities of the BYSMV P protein. Please see Figure 1—figure supplement 6.

In this Figure 4B, the CCR4^N^ terminus had no effect on BYSMV agMR replication, although the CCR4^N^ terminus could interact with the BYSMV P protein (Figure 1F). Thus, the direct CCR4-P interaction has no effect on the function of P protein. Moreover, the EEP domain is required for the positive effect of on BYSMV replication (Figure 4B, Figure 8, and the new Figure 6—figure supplement 1), indicating that the CCR4 deadenylation activity is essential for the CCR4 function in BYSMV infections.

The rhabdovirus P protein should form functional dimer structures in virus replication according to the review paper. Ivanov et al. (2011).

3) The deadenylation assays showed strong and direct evidence that P protein could tether CCR4 and that the N protein triggers degradation of RNA containing poly(A). However, the quality of Figure 8B should be improved; no significant increase of the degradation products can be seen in the figure at 15, 30, and 60 minutes. You should also describe the method and details how to quantify the deadenylated RNA in Figure 8B and C.

Thanks for your suggestions.

There is no increase of the degradation products at 15, 30, and 60 minutes, which should be resulted from the long reaction time. We have repeated the deadenylation assays in 10, 20, and 30 minutes. The new results clearly show significant increase of the degradation products. Please see the new Figure 8B and C.

To quantify the rates of deadenylated RNA, we fist quantified the 5’FAM-RNA band intensities by ImageJ software. The graphs plot the relative accumulation of remaining substrates estimated by electronic autoradiography in the gels of Figure 8D. The values of 5’FAM-RNA band intensities at 0 minutes were set to 100%. Data points represent mean values from three independent repeats. Error bars indicate standard errors of three independent experiments. The curve diagram was drown with GraphPad Prism. The detail method has been updated, please see subsection “Real-time quantitative RT-PCR (qRT-PCR)”.

4) RQC and RNA decay act as antiviral mechanisms, but they also play a role as endogenous suppressors of PTGS. This entails that HvCCR4 overexpression may promote viral replication through repressing PTGS. HvCCR4 is localized to viroplasm-like structures where viral mRNA is transcribed. In another scenario, it is possible that HvCCR4 degrades aberrant or excessive viral mRNAs to suppress antiviral RNA silencing, thereby promoting viral replication. Arabidopsis CCR4a is localized to both P-bodies and siRNA bodies (Moreno et al., 2013) and BYSMV infection promotes the generation of virus-derived siRNAs as shown in one of the authors' previous study. Do BYSMV P protein (or viroplasm-like structures) co-localize with siRNA bodies and HvCCR4? Is the accumulation of virus-derived siRNAs affected by HvCCR4 overexpression (Figure 6D and Figure 7F)? Please consider this possibility and discuss it in depth.

Thanks for these suggestions. RNA decay and RNA silencing have some relationship and the two pathways have some common complements such as AGO1. BYSMV GFP-P and SGS3-RFP were co-expressed in *N. benthamiana* leaves, and some but not all P-GFP bodies were adjacent to but not overlap with SGS-RFP bodies that is a siRNA body marker. These results indicate that some transient connection between BYSMV P bodies and siRNA bodies. Please see Figure 8—figure supplement 1A.

To evaluate whether HvCCR4 has suppressor activities, we used the classical co-infiltration assays to show that neither CCR4 RNAi nor CCR4 overexpression has effects on GFP-induced RNA silencing in *N. benthamiana* leaves. Please see Figure 8—figure supplement 1B and C.

The possibility of CCR4 effect in RNAi-mediated silencing will examined in our future studies. We also add some discussion, please see the Discussion section.

5) Figure 2D and E requires quantitative data showing how many foci were detected and how many were completely or adjacently co-localized in the absence/presence of P proteins.

Thanks, and we agree that quantitative data would provide solid conclusions.

We have done the quantitative analyses and clearly show that most the CCR4-GFP and DCP1-mCherry bodies (56 among 60 in total) are overlapped in the absence of the BYSMV P protein (N was co-expressed as a negative control, Figure 2E). When the P protein was co-expressed, 58 among 60 CCR4-GFP bodies were adjacent to DCP1-mCherry bodies. Please see the new Figure 2—figure supplement 2.

[Editors' note: further revisions were suggested prior to acceptance, as described below.]

The manuscript has been improved but there are some remaining issues that need to be addressed before acceptance, as outlined by reviewer 4. Could you please address these in the final revision?Reviewer #4:In this manuscript, Zhang et al. report that CCR4 RNA decay factor is hijacked by BYSMV P protein into viroplasm-like bodies to trigger degradation of cellular mRNAs bound by BYSMV N protein, allowing N protein to bind viral genomic RNA for viral replication. The findings deepen our understanding of the molecular mechanisms in host-virus arm race.In general, the conclusions supported by the data they presented and the data are of good quality. The authors also addressed most of the concerns raised during the last round of review. Listed below are my comments:1) The Title should be revised to reflect that CCR4 RNA decay factor is hijacked by BYSMV P.

Agreed, we have changed the Title as suggested as “CCR4, a RNA decay factor, is hijacked by a plant cytorhabdovirus phosphoprotein to facilitate virus replication”

2) CCR4 was identified to interact with BYSMV P protein through IP-MS, proteins identified in IP-MS should be listed in a table.

Agreed, we added a new supplementary file to list the P interacting proteins in IP-MS. Please new Supplementary file 1.

3) Figure 1—figure supplement 6A, labeling for the fifth lane is missing. The bottom western blot image only has 4 lanes, other images have 5 lanes.

Thanks, we changed the bottom image. The fifth lane is empty and cut out mistakenly. Please see the new Figure1—figure supplement 6A.

4) In Figure 6D-F, the authors show that barley plants infected with BYR-HvCCR4 developed more severe symptoms and have a higher level of BYSMV accumulation. Is it because that SBPH infected with BYR-HvCCR4 carries a higher amount of BYSMV?

Thanks for your suggestions. SBPH infected with BYR-HvCCR4 probably carry a higher virus amount than BYR-GUS. However, the SBPH with viruses were only incubated with healthy barley plants for a two-day inoculation, and then removed. The systemically infected leaves were collected for virus detection. Therefore, we propose that overexpression of CCR4 facilitate virus infections in systemically infected leaves.

In addition, accumulation of BYR-HvCCR4 is higher than BYR-GUS in infiltrated *N. benthamiana* leaves (Figure 6B and 6C). Collectively, these results clearly show that overexpression of CCR4 could facilitate virus infections in *N. benthamiana* and barley plants, as well as in planthopper vectors.

5) For all the western blot results, the loading control should be specified. For some of the IP results, for instance, Figure 1C and D, only the symbol "+" was used. The symbol "-" was missing above the images.

Changed. Rubisco complex large subunit (RbcL) detected by Stain-Free technology was used as a loading control. Please see updated Figure 1, Figure 4, Figure 6 and Figure 7.

The symbol "-" was added in the IP figures. Please see updated Figure 1, Figure 3, Figure 5 and Figure 7.

The writing should be improved. The authors should check throughout the manuscript for writing problems. Here are some examples.6) Introduction, "Thus, shortening of mRNA poly (A) tail..."The word "thus" should be removed.7) Introduction, "Thus, studying the interplay between.. . understanding the molecular mechanisms of host-pathogen interactions"The sentence should be removed.8) Subsection “BYSMV P interacts with barley HvCCR4 in vitro and in vivo”, "when N. benthamiana leaves... in contrast to the highly dynamic movement along ER tubules observed in the DMSO control"The sentence should be rephrased.9) Subsection “BYSMV P interacts with barley HvCCR4 in vitro and in vivo”, the results should be described at the end of the preceding paragraph.10) Subsection “HvCAF1 has a negative effect on BYSMV minigenome replication”, "Western blotting analyses revealed that.. .in different expression level of HvCAF1-Flag"The sentence should be rephrased.11) Subsection “HvCAF1 has a negative effect on BYSMV minigenome replication”, "thereby weakening the HvCCR4-P interactions that compromise HvCCR4 proviral functions in BYSMV replication"The sentence should be rephrased.12) Subsection “CCR4 triggers decay of the N-bound cellular mRNA”, "exhibit specificity improvement”.

Many thanks. All the writing errors have been changed or rephrased.